# TiAda: A Time-scale Adaptive Algorithm for Nonconvex Minimax Optimization

**Xiang Li,  Junchi Yang,  Niao He**
Department of Computer Science, ETH Zurich, Switzerland
{xiang.li,junchi.yang,niao.he}@inf.ethz.ch

## Abstract

Adaptive gradient methods have shown their ability to adjust the stepsizes on the fly in a parameter-agnostic manner, and empirically achieve faster convergence for solving minimization problems. When it comes to nonconvex minimax optimization, however, current convergence analyses of gradient descent ascent (GDA) combined with adaptive stepsizes require careful tuning of hyper-parameters and the knowledge of problem-dependent parameters. Such a discrepancy arises from the primal-dual nature of minimax problems and the necessity of delicate *time-scale separation* between the primal and dual updates in attaining convergence. In this work, we propose a *single-loop* adaptive GDA algorithm called TiAda for nonconvex minimax optimization that automatically adapts to the time-scale separation. Our algorithm is *fully parameter-agnostic* and can achieve *near-optimal complexities* simultaneously in deterministic and stochastic settings of nonconvex-strongly-concave minimax problems. The effectiveness of the proposed method is further justified numerically for a number of machine learning applications.

## 1 Introduction

Adaptive gradient methods, such as AdaGrad (Duchi et al., 2011), Adam (Kingma & Ba, 2015) and AMSGrad (Reddi et al., 2018), have become the default choice of optimization algorithms in many machine learning applications owing to their robustness to hyper-parameter selection and fast empirical convergence. These advantages are especially prominent in nonconvex regime with success in training deep neural networks (DNN). Classic analyses of gradient descent for smooth functions require the stepsize to be less than $2/l$, where $l$ is the smoothness parameter and often unknown for complicated models like DNN. Many adaptive schemes, usually with diminishing stepsizes based on cumulative gradient information, can adapt to such parameters and thus reducing the burden of hyper-parameter tuning (Ward et al., 2020; Xie et al., 2020). Such tuning-free algorithms are called *parameter-agnostic*, as they do not require any prior knowledge of problem-specific parameters, e.g., the smoothness or strong-convexity parameter.

In this work, we aim to bring the benefits of adaptive stepsizes to solving the following problem:

$$\min_{x\in\mathbb{R}^{d_1}} \max_{y\in\mathcal{Y}} f(x,y) = \mathbb{E}_{\xi\in P}\left[F(x,y;\xi)\right], \tag{1}$$

where $P$ is an unknown distribution from which we can drawn i.i.d. samples, $\mathcal{Y} \subset \mathbb{R}^{d_2}$ is closed and convex, and $f : \mathbb{R}^{d_1} \times \mathbb{R}^{d_2} \to \mathbb{R}$ is nonconvex in $x$. We call $x$ the primal variable and $y$ the dual variable. This minimax formulation has found vast applications in modern machine learning, notably generative adversarial networks (Goodfellow et al., 2014; Arjovsky et al., 2017), adversarial learning (Goodfellow et al., 2015; Miller et al., 2020), reinforcement learning (Dai et al., 2017; Modi et al., 2021), sharpness-aware minimization (Foret et al., 2021), domain-adversarial training (Ganin et al., 2016), etc. Albeit theoretically underexplored, adaptive methods are widely deployed in these applications in combination with popular minimax optimization algorithms such as (stochastic) gradient descent ascent (GDA), extragradient (EG) (Korpelevich, 1976), and optimistic GDA (Popov, 1980; Rakhlin & Sridharan, 2013); see, e.g., (Gulrajani et al., 2017; Daskalakis et al., 2018; Mishchenko et al., 2020; Reisizadeh et al., 2020), just to list a few.

While it seems natural to directly extend adaptive stepsizes to minimax optimization algorithms, a recent work by Yang et al. (2022a) pointed out that such schemes may not always converge without

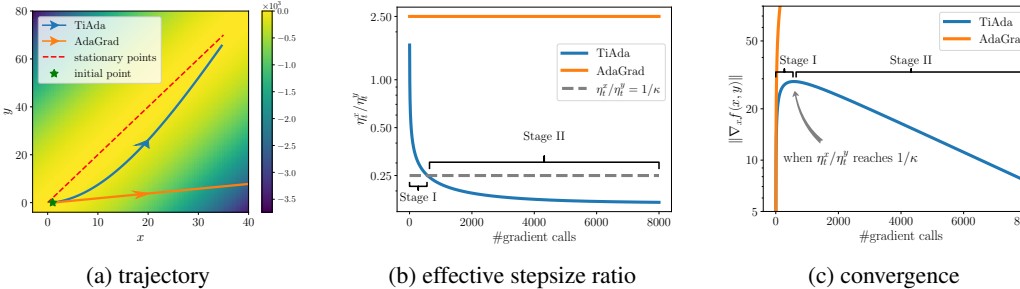

(a) trajectory           (b) effective stepsize ratio           (c) convergence

Figure 1: Comparison between TiAda and vanilla GDA with AdaGrad stepsizes (labeled as Ada-Grad) on the quadratic function (2) with $L = 2$ under a poor initial stepsize ratio, i.e., $\eta^x/\eta^y = 5$. Here, $\eta_t^x$ and $\eta_t^y$ are the effective stepsizes respectively for $x$ and $y$, and $\kappa$ is the condition number[1]. (a) shows the trajectory of the two algorithms and the background color demonstrates the function value $f(x, y)$. In (b), while the effective stepsize ratio stays unchanged for AdaGrad, TiAda adapts to the desired *time-scale separation* $1/\kappa$, which divides the training process into two stages. In (c), after entering Stage II, TiAda converges fast, whereas AdaGrad diverges.

knowing problem-dependent parameters. Unlike the case of minimization, convergent analyses of GDA and EG for nonconvex minimax optimization are subject to *time-scale separation* (Boţ & Böhm, 2020; Lin et al., 2020a; Sebbouh et al., 2022; Yang et al., 2022b) — the stepsize ratio of primal and dual variables needs to be smaller than a problem-dependent threshold — which is recently shown to be necessary even when the objective is strongly concave in $y$ with true gradients (Li et al., 2022). Moreover, Yang et al. (2022a) showed that GDA with standard adaptive stepsizes, that chooses the stepsize of each variable based only on the (moving) average of its own past gradients, fails to adapt to the time-scale separation requirement. Take the following nonconvex-strongly-concave function as a concrete example:

$$f(x, y) = -\frac{1}{2}y^2 + Lxy - \frac{L^2}{2}x^2, \tag{2}$$

where $L > 0$ is a constant. Yang et al. (2022a) proved that directly using adaptive stepsizes like AdaGrad, Adam and AMSGrad will fail to converge if the ratio of initial stepsizes of $x$ and $y$ (denoted as $\eta^x$ and $\eta^y$) is large. We illustrate this phenomenon in Figures 1(a) and 1(c), where AdaGrad diverges. To sum up, adaptive stepsizes designed for minimization, are not *time-scale adaptive* for minimax optimization and thus not *parameter-agnostic*.

To circumvent this time-scale separation bottleneck, Yang et al. (2022a) introduced an adaptive algorithm, NeAda, for problem (1) with nonconvex-strongly-concave objectives. NeAda is a two-loop algorithm built upon GDmax (Lin et al., 2020a) that after one primal variable update, updates the dual variable for multiple steps until a stopping criterion is satisfied in the inner loop. Although the algorithm is agnostic to the smoothness and strong-concavity parameters, there are several limitations that may undermine its performance in large-scale training: (a) In the stochastic setting, it gradually increases the number of inner loop steps ($k$ steps for the $k$-th outer loop) to improve the inner maximization problem accuracy, resulting in a possible waste of inner loop updates if the maximization problem is already well solved; (b) NeAda needs a large batchsize of order $\Omega\left(\epsilon^{-2}\right)$ to achieve the near-optimal convergence rate in theory; (c) It is not fully adaptive to the gradient noise, since it deploys different strategies for deterministic and stochastic settings.

In this work, we address all of the issues above by proposing TiAda (**Ti**me-scale **Ada**ptive Algorithm), a single-loop algorithm with time-scale adaptivity for minimax optimization. Specifically, one of our major modifications is setting the effective stepsize, i.e., the scale of (stochastic) gradient used in the updates, of the primal variable to the reciprocal of the *maximum* between the primal and dual variables' second moments, i.e., the sums of their past gradient norms. This ensures the effective stepsize ratio of $x$ and $y$ being upper bounded by a decreasing sequence, which eventually reaches the desired time-scale separation. Taking the test function (2) as an example, Figure 1 illustrates the time-scale adaptivity of TiAda: In Stage I, the stepsize ratio quickly decreases below the threshold; in Stage II, the ratio is stabilized and the gradient norm starts to converge fast.

We focus on the minimax optimization (1) that is strongly-concave in $y$, since other nonconvex regimes are far less understood even without adaptive stepsizes. Moreover, near stationary point

---

[1]Please refer to Section 2 for formal definitions of initial stepsize and effective stepsize. Note that the initial stepsize ratio, $\eta^x/\eta^y$, does not necessarily equal to the first effective stepsize ratio, $\eta_0^x/\eta_0^y$.

may not exist in nonconvex-nonconcave (NC-NC) problems and finding first-order local minimax point is already PPAD-complete (Daskalakis et al., 2021). We consider a constraint for the dual variable, which is common in convex optimization with adaptive stepsizes (Levy, 2017; Levy et al., 2018) and in the minimax optimization with non-adaptive stepsizes (Lin et al., 2020a). In summary, our contributions are as follows:

- We introduce the first *single-loop* and *fully parameter-agnostic* adaptive algorithm, TiAda, for nonconvex-strongly-concave (NC-SC) minimax optimization. It adapts to the necessary time-scale separation without large batchsize or any knowledge of problem-dependant parameters or target accuracy. TiAda finds an $\epsilon$-stationary point with an optimal complexity of $\mathcal{O}\left(\epsilon^{-2}\right)$ in the deterministic case, and a near-optimal sample complexity of $\mathcal{O}\left(\epsilon^{-(4+\delta)}\right)$ for any small $\delta > 0$ in the stochastic case. It shaves off the extra logarithmic terms in the complexity of NeAda with AdaGrad stepsize for both primal and dual variables (Yang et al., 2022a). TiAda is proven to be noise-adaptive, which is the first of its kind among nonconvex minimax optimization algorithms.

- While TiAda is based on AdaGrad stepsize, we generalize TiAda with other existing adaptive schemes, and conduct experiments on several tasks. The tasks include 1) test functions by Yang et al. (2022a) for showing the nonconvergence of GDA with adaptive schemes under poor initial stepsize ratios, 2) distributional robustness optimization (Sinha et al., 2018) on MNIST dataset with a NC-SC objective, and 3) training the NC-NC generative adversarial networks on CIFAR-10 dataset. In all tasks, we show that TiAda converges faster and is more robust compared with NeAda or GDA with other existing adaptive stepsizes.

## 1.1 Related Work

**Adaptive gradient methods.** AdaGrad brings about an adaptive mechanism for gradient-based optimization algorithm that adjusts its stepsize by keeping the averaged past gradients. The original AdaGrad was introduced for online convex optimization and maintains coordinate-wise stepsizes. In nonconvex stochastic optimization, AdaGrad-Norm with one learning rate for all directions is shown to achieve the same complexity as SGD (Ward et al., 2020; Li & Orabona, 2019), even with the high probability bound (Kavis et al., 2022; Li & Orabona, 2020). In comparison, RMSProp (Hinton et al., 2012) and Adam (Kingma & Ba, 2015) use the decaying moving average of past gradients, but may suffer from divergence (Reddi et al., 2018). Many variants of Adam are proposed, and a wide family of them, including AMSGrad, are provided with convergence guarantees (Zhou et al., 2018; Chen et al., 2018; Défossez et al., 2020; Zhang et al., 2022b). One of the distinguishing traits of adaptive algorithms is that they can achieve order-optimal rates without knowledge about the problem parameters, such as smoothness and variance of the noise, even in nonconvex optimization (Ward et al., 2020; Levy et al., 2021; Kavis et al., 2019).

**Adaptive minimax optimization algorithms.** The adaptive stepsize schemes are naturally extended to minimax optimization, both in theory and practice, notably in the training of GANs (Goodfellow, 2016; Gidel et al., 2018). In the convex-concave regime, several adaptive algorithms are designed based on EG and AdaGrad stepsize, and they inherit the parameter-agnostic characteristic (Bach & Levy, 2019; Antonakopoulos et al., 2019). In sharp contrast, when the objective function is nonconvex about one variable, most existing adaptive algorithms require knowledge of the problem parameters (Huang & Huang, 2021; Huang et al., 2021; Guo et al., 2021). Very recently, it was proved that a parameter-dependent ratio between two stepsizes is necessary for GDA in NC-SC minimax problems with non-adaptive stepsize (Li et al., 2022) and most existing adaptive stepsizes (Yang et al., 2022a). Heusel et al. (2017) shows the two-time-scaled GDA with non-adaptive stepsize or Adam will converge, but assuming the existence of an asymptotically stable attractor.

**Other NC-SC minimax optimization algorithms.** In the NC-SC setting, the most popular algorithms are GDA and GDmax, in which one primal variable update is followed by one or multiple steps of dual variable updates. Both of them can achieve $\mathcal{O}(\epsilon^{-2})$ complexity in the deterministic setting and $\mathcal{O}(\epsilon^{-4})$ sample complexity in the stochastic setting (Lin et al., 2020a; Chen et al., 2021; Nouiehed et al., 2019; Yang et al., 2020), which are not improvable in the dependency on $\epsilon$ given the existing lower complexity bounds (Zhang et al., 2021; Li et al., 2021). Later, several works further improved the dependency on the condition number with more complicated algorithms in deterministic (Yang et al., 2022b; Lin et al., 2020b) and stochastic settings (Zhang et al., 2022a). All of the algorithms above do not use adaptive stepsizes and rely on knowledge of the problem parameters.

## 1.2 NOTATIONS

We denote $l$ as the smoothness parameter, $\mu$ as the strong-concavity parameter, whose formal definitions will be introduced in Assumptions 3.1 and 3.2, and $\kappa := l/\mu$ as the condition number. We assume access to stochastic gradient oracle returning $[\nabla_x F(x, y; \xi), \nabla_y F(x, y; \xi)]$. For the minimax problem (1), we denote $y^*(x) := \arg\max_{y \in \mathcal{Y}} f(x, y)$ as the solution of the inner maximization problem, $\Phi(x) := f(x, y^*(x))$ as the primal function, and $\mathcal{P}_{\mathcal{Y}}(\cdot)$ as projection operator onto set $\mathcal{Y}$. For notational simplicity, we will use the name of an existing adaptive algorithm to refer to the simple combination of GDA and it, i.e., setting the stepsize of GDA to that adaptive scheme separately for both $x$ and $y$. For instance "AdaGrad" for minimax problems stands for the algorithm that uses AdaGrad stepsizes separately for $x$ and $y$ in GDA.

## 2 METHOD

We formally introduce the TiAda method in Algorithm 1, and the major difference with AdaGrad lies in line 5. Like AdaGrad, TiAda stores the accumulated squared (stochastic) gradient norm of the primal and dual variables in $v_t^x$ and $v_t^y$, respectively. We refer to hyper-parameters $\eta^x$ and $\eta^y$ as the *initial stepsizes*, and the actual stepsizes for updating in line 5 as *effective stepsizes* which are denoted by $\eta_t^x$ and $\eta_t^y$. TiAda adopts effective stepsizes $\eta_t^x = \eta^x / \max\left\{v_{t+1}^x, v_{t+1}^y\right\}^{\alpha}$ and $\eta_t^y = \eta^y / \left(v_{t+1}^y\right)^{\beta}$, while AdaGrad uses $\eta^x / \left(v_{t+1}^x\right)^{1/2}$ and $\eta^y / \left(v_{t+1}^y\right)^{1/2}$. In Section 3, our theoretical analysis suggests to choose $\alpha > 1/2 > \beta$. We will also illustrate in the next subsection that the $\max$ structure and different $\alpha, \beta$ make our algorithm adapt to the desired time-scale separation.

For simplicity of analysis, similar to AdaGrad-Norm (Ward et al., 2020), we use the norms of gradients for updating the effective stepsizes. A more practical coordinate-wise variant that can be used for high-dimensional models is presented in Section 4.1.

---

**Algorithm 1** TiAda (Time-scale Adaptive Algorithm)

1: **Input:** $(x_0, y_0)$, $v_0^x > 0$, $v_0^y > 0$, $\eta^x > 0$, $\eta^y > 0$, $\alpha > 0$, $\beta > 0$ and $\alpha > \beta$.
2: **for** $t = 0, 1, 2, ...$ **do**
3:     sample i.i.d. $\xi_t^x$ and $\xi_t^y$, and let $g_t^x = \nabla_x F(x_t, y_t; \xi_t^x)$ and $g_t^y = \nabla_y F(x_t, y_t; \xi_t^y)$
4:     $v_{t+1}^x = v_t^x + \|g_t^x\|^2$ and $v_{t+1}^y = v_t^y + \|g_t^y\|^2$
5:     $x_{t+1} = x_t - \frac{\eta^x}{\max\left\{v_{t+1}^x, v_{t+1}^y\right\}^{\alpha}} g_t^x$ and $y_{t+1} = \mathcal{P}_{\mathcal{Y}}\left(y_t + \frac{\eta^y}{\left(v_{t+1}^y\right)^{\beta}} g_t^y\right)$
6: **end for**

---

## 2.1 THE TIME-SCALE ADAPTIVITY OF TIADA

Current analyses of GDA with non-adaptive stepsizes require the time-scale, $\eta_t^x / \eta_t^y$, to be smaller than a threshold depending on problem constants such as the smoothness and the strong-concavity parameter (Lin et al., 2020a; Yang et al., 2022b). The intuition is that we should not aggressively update $x$ if the inner maximization problem has not yet been solved accurately, i.e., we have not found a good approximation of $y^*(x)$. Therefore, the effective stepsize of $x$ should be small compared with that of $y$. It is tempting to expect adaptive stepsizes to automatically find a suitable time-scale separation. However, the quadratic example (2) given by Yang et al. (2022a) shattered the illusion. In this example, the effective stepsize ratio stays the same along the run of existing adaptive algorithms, including AdaGrad (see Figure 1(b)), Adam and AMSGrad, and they fail to converge if the initial stepsizes are not carefully chosen (see Yang et al. (2022a) for details). As $v_t^x$ and $v_t^y$ only separately contain the gradients of $x$ and $y$, the effective stepsizes of two variables in these adaptive methods depend on their own history, which prevents them from cooperating to adjust the ratio.

Now we explain how TiAda adapts to both the required time-scale separation and small enough stepsizes. First, the ratio of our modified effective stepsizes is upper bounded by a decreasing sequence when $\alpha > \beta$:

$$\frac{\eta_t^x}{\eta_t^y} = \frac{\eta^x / \max\left\{v_{t+1}^x, v_{t+1}^y\right\}^{\alpha}}{\eta^y / \left(v_{t+1}^y\right)^{\beta}} \leq \frac{\eta^x / \left(v_{t+1}^y\right)^{\alpha}}{\eta^y / \left(v_{t+1}^y\right)^{\beta}} = \frac{\eta^x}{\eta^y \left(v_{t+1}^y\right)^{\alpha-\beta}}, \tag{3}$$

as $v_t^y$ is the sum of previous gradient norms and is increasing. Regardless of the initial stepsize ratio $\eta^x/\eta^y$, we expect the effective stepsize ratio to eventually drop below the desirable threshold for convergence. On the other hand, the effective stepsizes for the primal and dual variables are also upper bounded by decreasing sequences, $\eta^x/\left(v_{t+1}^x\right)^\alpha$ and $\eta^y/\left(v_{t+1}^y\right)^\beta$, respectively. Similar to AdaGrad, such adaptive stepsizes will reduce to small enough, e.g., $\mathcal{O}(1/l)$, to ensure convergence.

Another way to look at the effective stepsize of $x$ is

$$\eta_t^x = \frac{\eta^x}{\max\left\{v_{t+1}^x, v_{t+1}^y\right\}^\alpha} = \frac{\left(v_{t+1}^x\right)^\alpha}{\max\left\{v_{t+1}^x, v_{t+1}^y\right\}^\alpha} \cdot \frac{\eta^x}{\left(v_{t+1}^x\right)^\alpha}. \tag{4}$$

If the gradients of $y$ are small (i.e., $v_{t+1}^y < v_{t+1}^x$), meaning the inner maximization problem is well solved, then the first factor becomes 1 and the effective stepsize of $x$ is just the second factor, similar to the AdaGrad updates. If the term $v_{t+1}^y$ dominates over $v_{t+1}^x$, the first factor would be smaller than 1, allowing to slow down the update of $x$ and waiting for a better approximation of $y^*(x)$.

To demonstrate the time-scale adaptivity of TiAda, we conducted experiments on the quadratic mini-max example (2) with $L = 2$. As shown in Figure 1(b), while the effective stepsize ratio of AdaGrad stays unchanged for this particular function, TiAda progressively decreases the ratio. According to Lemma 2.1 of Yang et al. (2022a), $1/\kappa$ is the threshold where GDA starts to converge. We label the time period before reaching this threshold as Stage I, during which as shown in Figure 1(c), the gradient norm for TiAda increases. However, as soon as it enters Stage II, i.e., when the ratio drops below $1/\kappa$, TiAda converges fast to the stationary point. In contrast, since the stepsize ratio of AdaGrad never reaches this threshold, the gradient norm keeps growing.

## 3 THEORETICAL ANALYSIS OF TIADA

In this section, we study the convergence of TiAda under NC-SC setting with both deterministic and stochastic gradient oracles. We make the following assumptions to develop our convergence results.

**Assumption 3.1** (smoothness). *Function $f(x, y)$ is $l$-smooth ($l > 0$) in both $x$ and $y$, that is, for any $x_1, x_2 \in \mathbb{R}^{d_1}$ and $y_1, y_2 \in \mathcal{Y}$, we have*

$$\max\{\|\nabla_x f(x_1, y_1) - \nabla_x f(x_2, y_2)\|, \|\nabla_y f(x_1, y_1) - \nabla_y f(x_2, y_2)\|\} \leq l\left(\|x_1 - x_2\| + \|y_1 - y_2\|\right).$$

**Assumption 3.2** (strong-concavity in $y$). *Function $f(x, y)$ is $\mu$-strongly-concave ($\mu > 0$) in $y$, that is, for any $x \in \mathbb{R}^{d_1}$ and $y_1, y_2 \in \mathcal{Y}$, we have*

$$f(x, y_1) \geq f(x, y_2) + \langle \nabla_y f(x, y_1), y_1 - y_2 \rangle + \frac{\mu}{2}\|y_1 - y_2\|^2.$$

**Assumption 3.3** (interior optimal point). *For any $x \in \mathbb{R}^{d_1}$, $y^*(x)$ is in the interior of $\mathcal{Y}$.*

*Remark* 3.1. The last assumption ensures $\nabla_y f(x, y^*(x)) = 0$, which is important for AdaGrad-like stepsizes that use the sum of squared norms of past gradients in the denominator. If the gradient about $y$ is not 0 at $y^*(x)$, the stepsize will keep decreasing even near the optimal point, leading to slow convergence. This assumption could be potentially alleviated by using generalized AdaGrad stepsizes (Bach & Levy, 2019).

We aim to find a near stationary point for the minimax problem (1). Here, $(x, y)$ is defined to be an $\epsilon$ stationary point if $\|\nabla_x f(x, y)\| \leq \epsilon$ and $\|\nabla_y f(x, y)\| \leq \epsilon$ in the deterministic setting, or $\mathbb{E}\|\nabla_x f(x, y)\|^2 \leq \epsilon^2$ and $\mathbb{E}\|\nabla_y f(x, y)\|^2 \leq \epsilon^2$ in the stochastic setting, where the expectation is taken over all the randomness in the algorithm. This stationarity notion can be easily translated to the near-stationarity of the primal function $\Phi(x) = \max_{y \in \mathcal{Y}}(x, y)$ (Yang et al., 2022b). Under our analyses, TiAda is able to achieve the optimal $\mathcal{O}\left(\epsilon^{-2}\right)$ complexity in the deterministic setting and a near-optimal $\mathcal{O}\left(\epsilon^{-(4+\delta)}\right)$ sample complexity for any small $\delta > 0$ in the stochastic setting.

### 3.1 DETERMINISTIC SETTING

In this subsection, we assume to have access to the exact gradients of $f(\cdot, \cdot)$, and therefore we can replace $\nabla_x F(x_t, y_t; \xi_t^x)$ and $\nabla_y F(x_t, y_t; \xi_t^y)$ by $\nabla_x f(x_t, y_t)$ and $\nabla_y f(x_t, y_t)$ in Algorithm 1.

**Theorem 3.1** (deterministic setting). *Under Assumptions 3.1 to 3.3, Algorithm 1 with deterministic gradient oracles satisfies that for any $0 < \beta < \alpha < 1$, after $T$ iterations,*

$$\frac{1}{T} \sum_{t=0}^{T-1} \|\nabla_x f(x_t, y_t)\|^2 + \frac{1}{T} \sum_{t=0}^{T-1} \|\nabla_y f(x_t, y_t)\|^2 \leq \mathcal{O}\left(\frac{1}{T}\right).$$

This theorem implies that for any initial stepsizes, TiAda finds an $\epsilon$-stationary point within $\mathcal{O}(\epsilon^{-2})$ iterations. Such complexity is comparable to that of nonadaptive methods, such as vanilla GDA (Lin et al., 2020a), and is optimal in the dependency of $\epsilon$ (Zhang et al., 2021). Like NeAda (Yang et al., 2022a), TiAda does not need any prior knowledge about $\mu$ and $l$, but it improves over NeAda by removing the logarithmic term in the complexity. Notably, we provide a unified analysis for a wide range of $\alpha$ and $\beta$, while most existing literature on AdaGrad-like stepsizes only validates a specific hyper-parameter, e.g., $\alpha = 1/2$ in minimization problems (Ward et al., 2020; Kavis et al., 2019).

## 3.2 Stochastic Setting

In this subsection, we assume the access to a stochastic gradient oracle, that returns unbiased noisy gradients, $\nabla_x F(x, y; \xi)$ and $\nabla_y F(x, y; \xi)$. Also, we make the following additional assumptions.

**Assumption 3.4** (stochastic gradients). *For $z \in \{x, y\}$, we have $\mathbb{E}_\xi [\nabla_z F(x, y, \xi)] = \nabla_z f(x, y)$. In addition, there exists a constant $G$ such that $\|\nabla_z F(x, y, \xi)\| \leq G$ for any $x \in \mathbb{R}^{d_1}$ and $y \in \mathcal{Y}$.*

**Assumption 3.5** (bounded primal function value). *There exists a constant $\Phi_{\max} \in \mathbb{R}$ such that for any $x \in \mathbb{R}^{d_1}$, $\Phi(x)$ is upper bounded by $\Phi_{\max}$.*

*Remark 3.2.* The bounded gradients and function value are assumed in many works on adaptive algorithms (Kavis et al., 2022; Levy et al., 2021). This implies the domain of $y$ is bounded, which is also assumed in the analyses of AdaGrad (Levy, 2017; Levy et al., 2018). In neural networks with rectified activations, because of its scale-invariance property (Dinh et al., 2017), imposing boundedness of $y$ does not affect the expressiveness. Wasserstein GANs (Arjovsky et al., 2017) also use projections on the critic to restrain the weights on a small cube around the origin.

**Assumption 3.6** (second order Lipschitz continuity for $y$). *For any $x_1, x_2 \in \mathbb{R}^{d_1}$ and $y_1, y_2 \in \mathcal{Y}$, there exists constant $L$ such that $\left\|\nabla_{xy}^2 f(x_1, y_1) - \nabla_{xy}^2 f(x_2, y_2)\right\| \leq L\left(\|x_1 - x_2\| + \|y_1 - y_2\|\right)$ and $\left\|\nabla_{yy}^2 f(x_1, y_1) - \nabla_{yy}^2 f(x_2, y_2)\right\| \leq L\left(\|x_1 - x_2\| + \|y_1 - y_2\|\right)$.*

*Remark 3.3.* Chen et al. (2021) also impose this assumption to achieve the optimal $\mathcal{O}\left(\epsilon^{-4}\right)$ complexity for GDA with non-adaptive stepsizes for solving NC-SC minimax problems. Together with Assumption 3.3, we can show that $y^*(\cdot)$ is smooth. Nevertheless, without this assumption, Lin et al. (2020a) only show a worse complexity of $\mathcal{O}\left(\epsilon^{-5}\right)$ for GDA without large batchsize.

**Theorem 3.2** (stochastic setting). *Under Assumptions 3.1 to 3.6, Algorithm 1 with stochastic gradient oracles satisfies that for any $0 < \beta < \alpha < 1$, after $T$ iterations,*

$$\frac{1}{T}\mathbb{E}\left[\sum_{t=0}^{T-1} \|\nabla_x f(x_t, y_t)\|^2 + \sum_{t=0}^{T-1} \|\nabla_y f(x_t, y_t)\|^2\right] \leq \mathcal{O}\left(T^{\alpha-1} + T^{-\alpha} + T^{\beta-1} + T^{-\beta}\right).$$

TiAda can achieve the complexity arbitrarily close to the optimal sample complexity, $\mathcal{O}\left(\epsilon^{-4}\right)$ (Li et al., 2021), by choosing $\alpha$ and $\beta$ arbitrarily close to 0.5. Specifically, TiAda achieves a complexity of $\mathcal{O}\left(\epsilon^{-(4+\delta)}\right)$ for any small $\delta > 0$ if we set $\alpha = 0.5 + \delta/(8 + 2\delta)$ and $\beta = 0.5 - \delta/(8 + 2\delta)$. Notably, this matches the complexity of NeAda with AdaGrad stepsizes for both variables (Yang et al., 2022a). NeAda may attain $\widetilde{\mathcal{O}}(\epsilon^{-4})$ complexity with more complicated subroutines for $y$.

Theorem 3.2 implies that TiAda is fully agnostic to problem parameters, e.g., $\mu$, $l$ and $\sigma$. GDA with non-adaptive stepsizes (Lin et al., 2020a) and vanilla single-loop adaptive methods (Huang & Huang, 2021), such as AdaGrad and AMSGrad, all require knowledge of these parameters. Compared with the only parameter-agnostic algorithm, NeAda, our algorithm has several advantages. First, TiAda is a single-loop algorithm, while NeAda (Yang et al., 2022a) needs increasing inner-loop steps and a huge batchsize of order $\Omega\left(\epsilon^{-2}\right)$ to achieve its best complexity. Second, our stationary guarantee is for $\mathbb{E}\|\nabla_x f(x, y)\|^2 \leq \epsilon^2$, which is stronger than $\mathbb{E}\|\nabla_x f(x, y)\| \leq \epsilon$ guarantee in NeAda. Last but not least, although NeAda does not need to know the exact value of variance $\sigma$ in the stochastic setting when $\sigma > 0$, NeAda uses a different stopping criterion for the inner loop in the deterministic

setting when $\sigma = 0$, so it still needs partial information about $\sigma$. In comparison, TiAda achieves the (near) optimal complexity in both settings with the same strategy.

Consistent with the intuition of time-scale adaptivity in Section 2.1, the convergence result can be derived in two stages. In Stage I, according to the upper bound of the ratio in Equation (3), we expect the term $1/\left(v_{t+1}^y\right)^{\alpha-\beta}$ reduces to a constant $c$, a desirable time-scale separation. This means that $v_{t+1}^y$ has to grow to nearly $(1/c)^{1/(\alpha-\beta)}$. In Stage II, when the time-scale separation is satisfied, TiAda converges at a speed specified in Theorem 3.2. This indicates that the proximity between $\alpha$ and $\beta$ affects the speed trade-off between Stage I and II. When $\alpha$ and $\beta$ are close, we have a faster overall convergence rate close to the optimality, but suffer from a longer transition phase in Stage I, albeit by only a constant term. We also present an empirical ablation study on the convergence behavior with different choices of $\alpha$ and $\beta$ in Appendix A.2.

*Remark* 3.4. In TiAda, the update of $x$ requires to know the gradients of $y$ (or $v_{t+1}^y$). However, in some applications that concern about privacy, one player might not access the information about the other player (Koller & Pfeffer, 1995; Foster & Young, 2006; He et al., 2016). Therefore, we also consider a variant of TiAda without taking the maximum of gradient norms, i.e., setting the effective stepsize of $x$ in Algorithm 1 to $\eta^x/\left(v_{t+1}^x\right)^\alpha$. This variant achieves a sub-optimal complexity of $\widetilde{\mathcal{O}}\left(\epsilon^{-6}\right)$. This result further justifies the importance of coordination between adaptive stepsizes of two players for achieving faster convergence in minimax optimization. The algorithm and convergence results are presented in Appendix C.4.

## 4 Experiments

In this section, we first present extensions of TiAda that accommodate other adaptive schemes besides AdaGrad and are more practical in deep models. Then we present empirical results of TiAda and compare it with (i) simple combinations of GDA and adaptive stepsizes, which are commonly used in practice, and (ii) NeAda with different adaptive mechanisms (Yang et al., 2022a). Our experiments include test functions proposed by Yang et al. (2022a), the NC-SC distributional robustness optimization (Sinha et al., 2018), and training the NC-NC Wasserstein GAN with gradient penalty (Gulrajani et al., 2017). We believe that this not only validates our theoretical results but also shows the potential of our algorithm in real-world scenarios. To show the strength of being parameter-agnostic of TiAda, in all the experiments, we merely select $\alpha = 0.6$ and $\beta = 0.4$ without further tuning those two hyper-parameters. All experimental details including the neural network structure and hyper-parameters are described in Appendix A.1.

### 4.1 Extensions to Other Adaptive Stepsizes and High-dimensional Models

Although we design TiAda upon AdaGrad-Norm, it is easy and intuitive to apply other adaptive schemes like Adam and AMSGrad. To do so, for $z \in \{x, y\}$, we replace the definition of $g_t^z$ and $v_{t+1}^z$ in line 3 and 4 of Algorithm 1 to

$$g_t^z = \beta_t^z g_{t-1}^z + (1 - \beta_t^z)\nabla_z F(x_t, y_t; \xi_t^z), \quad v_{t+1}^z = \psi\left(v_0, \left\{\|\nabla_z F(x_i, y_i; \xi_i^z)\|^2\right\}_{i=0}^t\right),$$

where $\{\beta_t^z\}$ is the momentum parameters and $\psi$ is the second moment function. Some common stepsizes that fit in this generalized framework can be seen in Table 1 in the appendix. Since Adam is widely used in many deep learning tasks, we also implement generalized TiAda with Adam stepsizes in our experiments for real-world applications, and we label it "TiAda-Adam".

Besides generalizing TiAda to accommodate different stepsize schemes, for high-dimensional models, we also provide a coordinate-wise version of TiAda. Note that we cannot simply change everything in Algorithm 1 to be coordinate-wise, because we use the gradients of $y$ in the stepsize of $x$ and there are no corresponding relationships between the coordinates of $x$ and $y$. Therefore, in light of our intuition in Equation (4), we use the global accumulated gradient norms to dynamically adjust the stepsize of $x$. Denote the second moment (analogous to $v_{t+1}^x$ in Algorithm 1) for the $i$-th coordinate of $x$ at the $t$-th step as $v_{t+1,i}^x$ and globally $v_{t+1}^x := \sum_{i=1}^{d_1} v_{t+1,i}^x$. We also use similar notations for $y$. Then, the update for the $i$-th parameter, i.e., $x^i$ and $y^i$, can be written as

$$\begin{cases} x_{t+1}^i = x_t^i - \frac{\left(v_{t+1}^x\right)^\alpha}{\max\{v_{t+1}^x, v_{t+1}^y\}^\alpha} \cdot \frac{\eta^x}{\left(v_{t+1,i}^x\right)^\alpha}\nabla_{x^i} f(x_t, y_t) \\ y_{t+1}^i = y_t^i + \frac{\eta^y}{\left(v_{t+1,i}^y\right)^\beta}\nabla_{y^i} f(x_t, y_t). \end{cases}$$

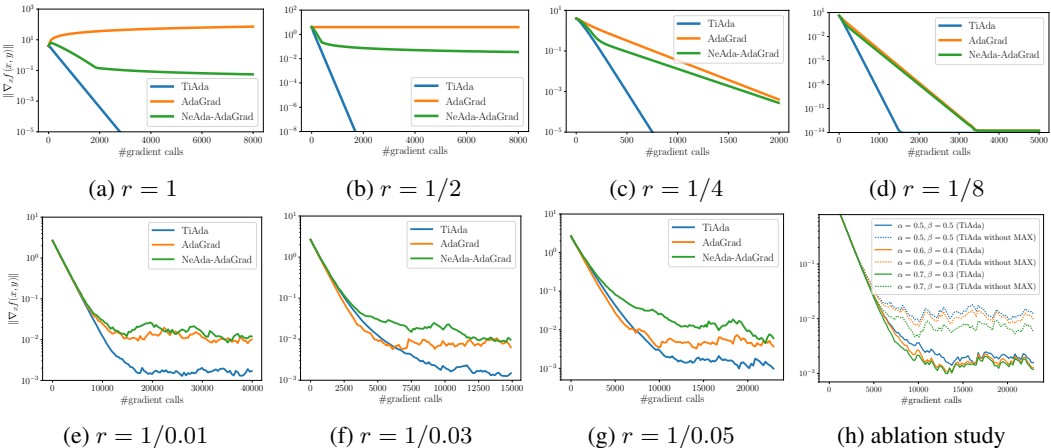

Figure 2: Comparison of algorithms on test functions. $r = \eta^x/\eta^y$ is the initial stepsize ratio. In the first row, we use the quadratic function (2) with $L = 2$ under deterministic gradient oracles. For the second row, we test the methods on the McCormick function with noisy gradients.

Our results in the following subsections provide strong empirical evidence for the effectiveness of these TiAda variants, and developing convergence guarantees for them would be an interesting future work. We believe our proof techniques for TiAda, together with existing convergence results for coordinate-wise AdaGrad and AMSGrad (Zhou et al., 2018; Chen et al., 2018; Défossez et al., 2020), can shed light on the theoretical analyses of these variants.

## 4.2 TEST FUNCTIONS

Firstly, we examine TiAda on the quadratic function (2) that shows the non-convergence of simple combinations of GDA and adaptive stepsizes (Yang et al., 2022a). Since our TiAda is based on AdaGrad, we compare it to GDA with AdaGrad stepsize and NeAda-AdaGrad (Yang et al., 2022a). The results are shown in the first row of Figure 2. When the initial ratio is poor, TiAda and NeAda-AdaGrad always converge while AdaGrad diverges. NeAda also suffers from slow convergence when the initial ratio is poor, e.g., 1 and 1/2 after 2000 iterations. In contrast, TiAda automatically balances the stepsizes and converges fast under all ratios.

For the stochastic case, we follow Yang et al. (2022a) and conduct experiments on the McCormick function which is more complicated and 2-dimensional: $f(x, y) = \sin(x_1 + x_2) + (x_1 - x_2)^2 - \frac{3}{2}x_1 + \frac{5}{2}x_2 + 1 + x_1y_1 + x_2y_2 - \frac{1}{2}(y_1^2 + y_2^2)$. TiAda consistently outperforms AdaGrad and NeAda-AdaGrad as demonstrated in the second row of Figure 2 regardless of the initial ratio. In this function, we also run an ablation study on the effect of our design that uses max-operator in the update of $x$. We compare TiAda with and its variant without the max-operator, TiAda without MAX (Algorithm 2 in the appendix) whose effective stepsizes of $x$ are $\eta^x/(v_{t+1}^x)^\alpha$. According to Figure 2(h), TiAda converges to smaller gradient norms under all configurations of $\alpha$ and $\beta$.

## 4.3 DISTRIBUTIONAL ROBUSTNESS OPTIMIZATION

In this subsection, we consider the distributional robustness optimization (Sinha et al., 2018). We target training the model weights, the primal variable $x$, to be robust to the perturbations in the image inputs, the dual variable $y$. The problem can be formulated as:

$$\min_x \max_{y=[y_1,\dots,y_n]} \frac{1}{n}\sum_{i=1}^{n} f_i(x, y_i) - \gamma\|y_i - v_i\|^2, \tag{5}$$

where $f_i$ is the loss function of the $i$-th sample, $v_i$ is the $i$-th input image, and $y_i$ is the corresponding perturbation. There are a total of $n$ samples and $\gamma$ is a trade-off hyper-parameter between the original loss and the penalty of the perturbations. If $\gamma$ is large enough, the problem is NC-SC.

We conduct the experiments on the MNIST dataset (LeCun, 1998). In the left two plots of Figure 3, we compare TiAda with AdaGrad and NeAda-AdaGrad in terms of convergence. Since it is common in practice to update $y$ 15 times after each $x$ update (Sinha et al., 2018) for better generalization error,

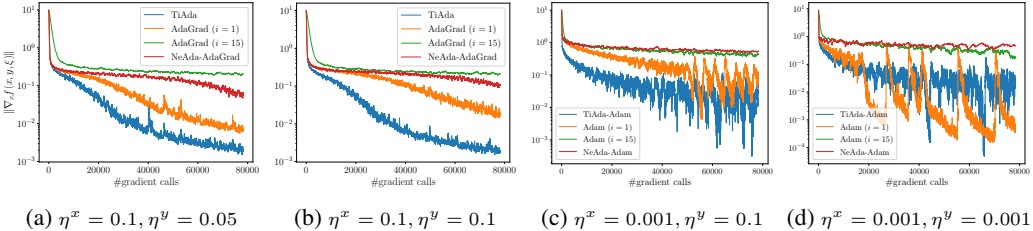

(a) $\eta^x = 0.1, \eta^y = 0.05$ (b) $\eta^x = 0.1, \eta^y = 0.1$ (c) $\eta^x = 0.001, \eta^y = 0.1$ (d) $\eta^x = 0.001, \eta^y = 0.001$

Figure 3: Comparison of the algorithms on distributional robustness optimization (5). We use $i$ in the legend to indicate the number of inner loops. Here we present two sets of stepsize configurations for the comparisons of AdaGrad-like and Adam-like algorithms. Please refer to Appendix A.3 for extensive experiments on larger ranges of stepsizes, and it will be shown that TiAda is the best among all stepsize combinations in our grid.

we implement AdaGrad using both single and 15 iterations of inner loop (update of $y$). In order to show that TiAda is more robust to the initial stepsize ratio, we compare two sets of initial stepsize configurations with two different ratios. In both cases, TiAda outperforms NeAda and AdaGrad, especially when $\eta^x = \eta^y = 0.1$, the performance gap is large. In the right two plots of Figure 3, the Adam variants are compared. In this case, we find that TiAda is not only faster, but also more stable comparing to Adam with one inner loop iteration.

## 4.4 GENERATIVE ADVERSARIAL NETWORKS

Another successful and popular application of minimax optimization is generative adversarial networks. In this task, a discriminator (or critic) is trained to distinguish whether an image is from the dataset. At the same time, a generator is mutually trained to synthesize samples with the same distribution as the training dataset so as to fool the discriminator. We use WGAN-GP loss (Gulrajani et al., 2017), which imposes the discriminator to be a 1-Lipschitz function, with CIFAR-10 dataset (Krizhevsky et al., 2009) in our experiments.

Since TiAda is a single-loop algorithm, for fair comparisons, we also update the discriminator only once for each generator update in Adam. In Figure 4, we plot the inception scores (Salimans et al., 2016) of TiAda-Adam

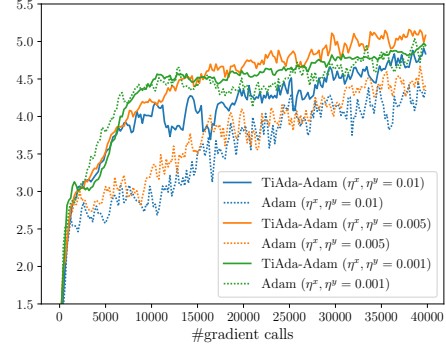

Figure 4: Inception score on WGAN-GP.

and Adam under different initial stepsizes. We use the same color for the same initial stepsizes, and different line styles to distinguish the two methods, i.e., solid lines for TiAda-Adam and dashed lines for Adam. For all the three initial stepsizes we consider, TiAda-Adam achieves higher inception scores. Also, TiAda-Adam is more robust to initial stepsize selection, as the gap between different solid lines at the end of training is smaller than the dashed lines.

## 5 CONCLUSION

In this work, we bring in adaptive stepsizes to nonconvex minimax problems in a parameter-agnostic manner. We designed the first time-scale adaptive algorithm, TiAda, which progressively adjusts the effective stepsize ratio and reaches the desired time-scale separation. TiAda is also noise adaptive and does not require large batchsizes compared with the existing parameter-agnostic algorithm for nonconvex minimax optimization. Furthermore, TiAda is able to achieve optimal and near-optimal complexities respectively wtih deterministic and stochastic gradient oracles. We also empirically showcased the advantages of TiAda over NeAda and GDA with adaptive stepsizes on several tasks, including simple test functions, as well as NC-SC and NC-NC real-world applications. It remains an interesting problem to study whether TiAda can escape stationary points that are not local optimum, like adaptive methods for minimization problems (Staib et al., 2019).

ACKNOWLEDGMENT

We thank Dr. Anas Barakat and anonymous reviewers for their valuable feedback. The work is supported by ETH research grant and Swiss National Science Foundation (SNSF) Project Funding No. 200021-207343.

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

# A    SUPPLEMENTARY TO EXPERIMENTS

Table 1: Stepsize schemes fit in generalized TiAda. See also Yang et al. (2022a).

| Algorithms | first moment parameter $\beta_t$ | second moment function $\psi\left(v_0, \{u_i^2\}_{i=0}^t\right)$ |
|---|---|---|
| AdaGrad (TiAda) | $\beta_t = 0$ | $v_0 + \sum_{i=0}^t u_i^2$ |
| GDA | $\beta_t = 0$ | $1$ |
| Adam | $0 < \beta_t < 1$ | $\gamma^{t+1} v_0 + (1 - \gamma) \sum_{i=0}^t \gamma^{t-i} u_i^2$ |
| AMSGrad | $0 < \beta_t < 1$ | $\max_{m=0,\ldots,t} \gamma^{m+1} v_0 + (1 - \gamma) \sum_{i=0}^m \gamma^{m-i} u_i^2$ |

## A.1    EXPERIMENTAL DETAILS

In this section, we will summarize the experimental settings and hyper-parameters used. As we mentioned, since we try to develop a parameter-agnostic algorithm without tuning the hyper-parameters much, if not specified, we simply use $\alpha = 0.6$ and $\beta = 0.4$ for all experiments. For fair comparisons, we used the same hyper-parameters when comparing our TiAda with other algorithms.

**Test Functions**    For Figure 1 and the first row of Figure 2, we conduct experiments on problem (2) with $L = 2$. We use initial stepsize $\eta^y = 0.2$ and initial point $(1, 0.01)$ for all runs. As for the McCormick function used in the second row of Figure 2, we chose $\eta^y = 0.01$, and the noises added to the gradients are from zero-mean Gaussian distribution with variance $0.01$.

**Distributional Robustness Optimization**    For results shown in Figures 3, 6 and 7, we adapt code from Lv (2019), and used the same hyper-parameter setting as Sinha et al. (2018); Sebbouh et al. (2022), i.e., $\gamma = 1.3$. The model we used is a three layer convolutional neural network (CNN) with a final fully-connected layer. For each layer, batch normalization and ELU activation are used. The width of each layer is $(32, 64, 128, 512)$. The setting is the same as Sinha et al. (2018); Yang et al. (2022a). We set the batchsize as $128$, and for the Adam-like optimizers, including Adam, NeAda-Adam and TiAda-Adam, we use $\beta_1 = 0.9, \beta_2 = 0.999$ for the first moment and second moment parameters.

**Generative Adversarial Networks** For this part, we use the code adapted from Green9 (2018). To produce the results in Figure 4, a four layer CNN and a four layer CNN with transpose convolution layers are used respectively for the discriminator and generator. Following a similar setting as Daskalakis et al. (2018), we set batchsize as $512$, the dimension of latent variable as $50$ and the weight of gradient penalty term as $10^{-4}$. For the Adam-like optimizers, we set $\beta_1 = 0.5, \beta_2 = 0.9$. To get the inception score, we feed the pre-trained inception network with 8000 synthesized samples.

## A.2 ABLATION STUDY ON CONVERGENCE BEHAVIOR WITH DIFFERENT $\alpha$ AND $\beta$

We conduct experiments on the quadratic minimax problem (2) with $L = 2$ to study the effect of hyper-parameters $\alpha$ and $\beta$ on the convergence behavior of TiAda. As discussed in Sections 1 and 3.2, we refer to the period before the stepsize ratio reduce to the convergence threshold as Stage I, and the period after that as Stage II. In order to accentuate the difference between these two stages, we pick a large initial stepsize ratio $\eta^x/\eta^y = 20$. We compare 4 different pairs of $\alpha$ and $\beta$: $\alpha \in \{0.59, 0.6, 0.61, 0.62\}$ and $\beta = 1-\alpha$. From Figure 5, we observed that as soon as TiAda enters Stage II, the norm of gradients start to drop. Moreover, the closer $\alpha$ and $\beta$ are to 0.5, the more time TiAda remains in Stage I, which confirms the intuitions behind our analysis in Section 3.2.

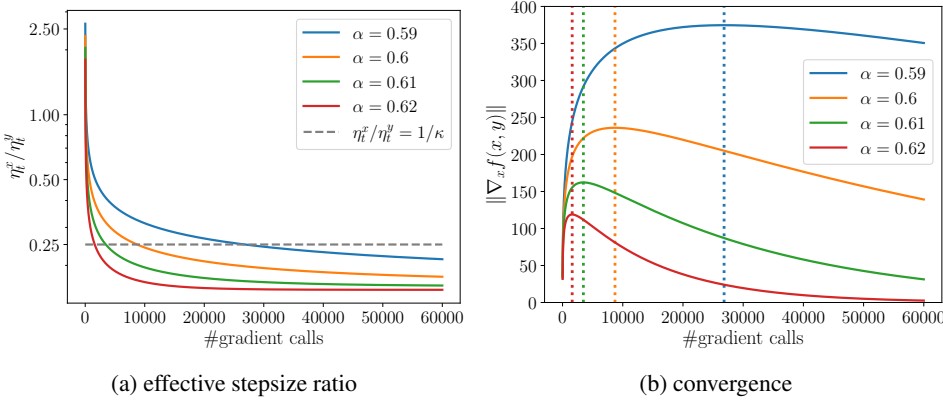

(a) effective stepsize ratio         (b) convergence

Figure 5: Illustration of the effect of $\alpha$ and $\beta$ on the two stages in TiAda's time-scale adaptation process. We set $\beta = 1 - \alpha$. The dashed line on the right plot represents the first iteration when the effective stepsize ratio is below $1/\kappa$.

## A.3 ADDITIONAL EXPERIMENTS ON DISTRIBUTIONAL ROBUSTNESS OPTIMIZATION

We use a grid of stepsize combinations to evaluate TiAda and compare it with NeAda and GDA with corresponding adaptive stepsizes. For AdaGrad-like algorithms, we use $\{0.1, 0.05, 0.01, 0.0005\}$ for both $\eta^x$ and $\eta^y$, and the results are reported in Figure 6. For Adam-like algorithms, we use $\{0.001, 0.0005, 0.0001\}$ for $\eta^x$ and $\{0.1, 0.05, 0.005, 0.001\}$ for $\eta^y$, and the results are shown in Figure 7. We note that since Adam uses the reciprocal of the moving average of gradient norms, it is extremely unstable when the gradients are small. Therefore, Adam-like algorithms often experience instability when they are near stationary points.

## B HELPER LEMMAS

**Lemma B.1** (Lemma A.2 in Yang et al. (2022a)). *Let $x_1, ..., x_T$ be a sequence of non-negative real numbers, $x_1 > 0$ and $0 < \alpha < 1$. Then we have*

$$\left(\sum_{t=1}^{T} x_t\right)^{1-\alpha} \leq \sum_{t=1}^{T} \frac{x_t}{\left(\sum_{k=1}^{t} x_k\right)^{\alpha}} \leq \frac{1}{1-\alpha}\left(\sum_{t=1}^{T} x_t\right)^{1-\alpha}.$$

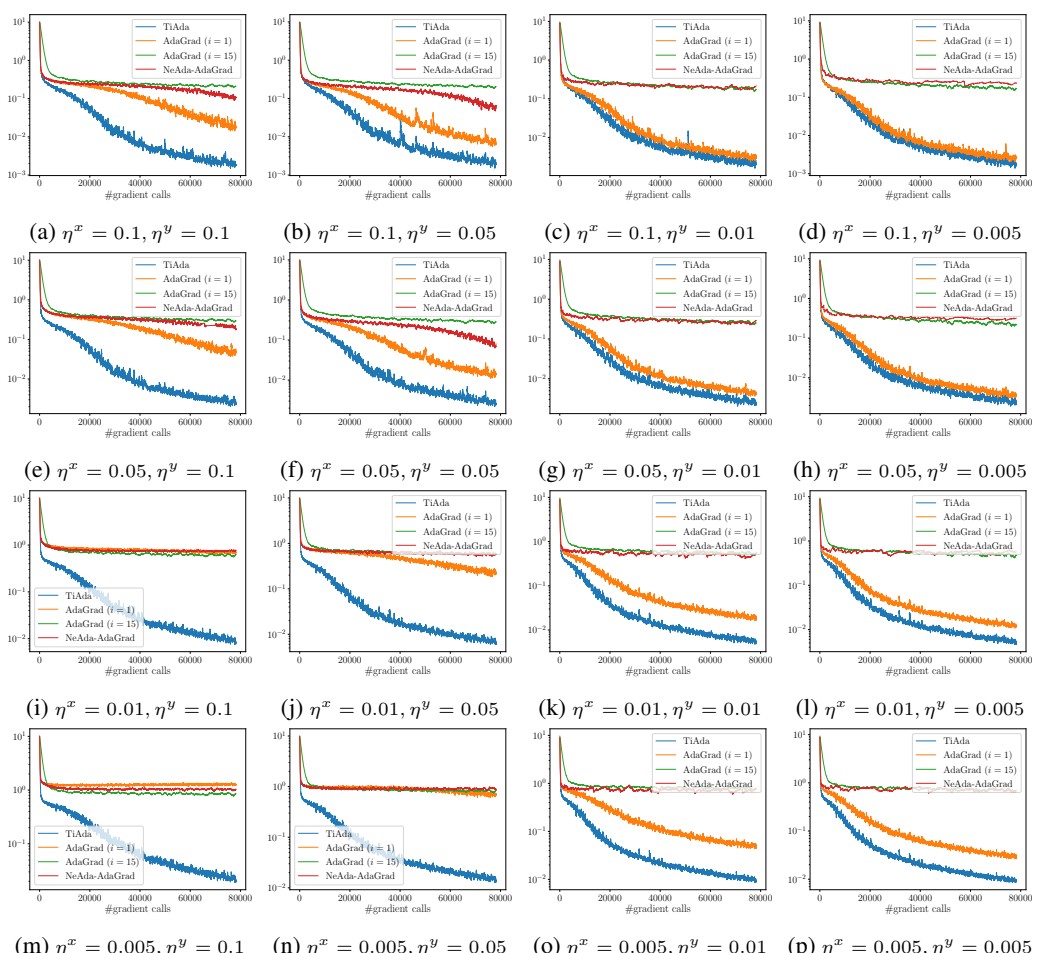

Figure 6: Gradient norms in $x$ of AdaGrad-like algorithms on distributional robustness optimization (5). We use $i$ in the legend to indicate the number of inner loops.

*When $\alpha = 1$, we have*

$$\sum_{t=1}^{T} \frac{x_t}{\left(\sum_{k=1}^{t} x_k\right)^{\alpha}} \leq 1 + \log\left(\frac{\sum_{t=1}^{t} x_t}{x_1}\right).$$

**Lemma B.2** (smoothness of $\Phi(\cdot)$ and Lipschitzness of $y^*(\cdot)$. Lemma 4.3 in Lin et al. (2020a)). *Under Assumptions 3.1 and 3.2, we have $\Phi(\cdot)$ is $(l + \kappa l)$-smooth with $\nabla\Phi(x) = \nabla_x f(x, y^*(x))$, and $y^*(\cdot)$ is $\kappa$-Lipschitz.*

**Lemma B.3** (smoothness of $y^*(\cdot)$. Lemma 2 in Chen et al. (2021)). *Under Assumptions 3.1, 3.2 and 3.6, we have that with $\widehat{L} = \frac{L + L\kappa}{\mu} + \frac{l(L + L\kappa)}{\mu^2}$,*

$$\|\nabla y^*(x_1) - \nabla y^*(x_2)\| \leq \widehat{L}\|x_1 - x_2\|.$$

## C    PROOFS

For notational convenience in the proofs, we denote the stochastic gradient as $\nabla_x \widetilde{f}(x_t, y_t)$ and $\nabla_y \widetilde{f}(x_t, y_t)$. Also denote $y_t^* = y^*(x_t)$, $\eta_t = \frac{\eta^x}{\max\{v_{t+1}^x, v_{t+1}^y\}^{\alpha}}$, $\gamma_t = \frac{\eta^y}{\left(v_{t+1}^y\right)^{\beta}}$, $\Phi^* = \min_{x \in \mathbb{R}^{d_1}} \Phi(x)$, and $\Delta\Phi = \Phi_{\max} - \Phi^*$. We use $\mathbf{1}$ as the indicator function.

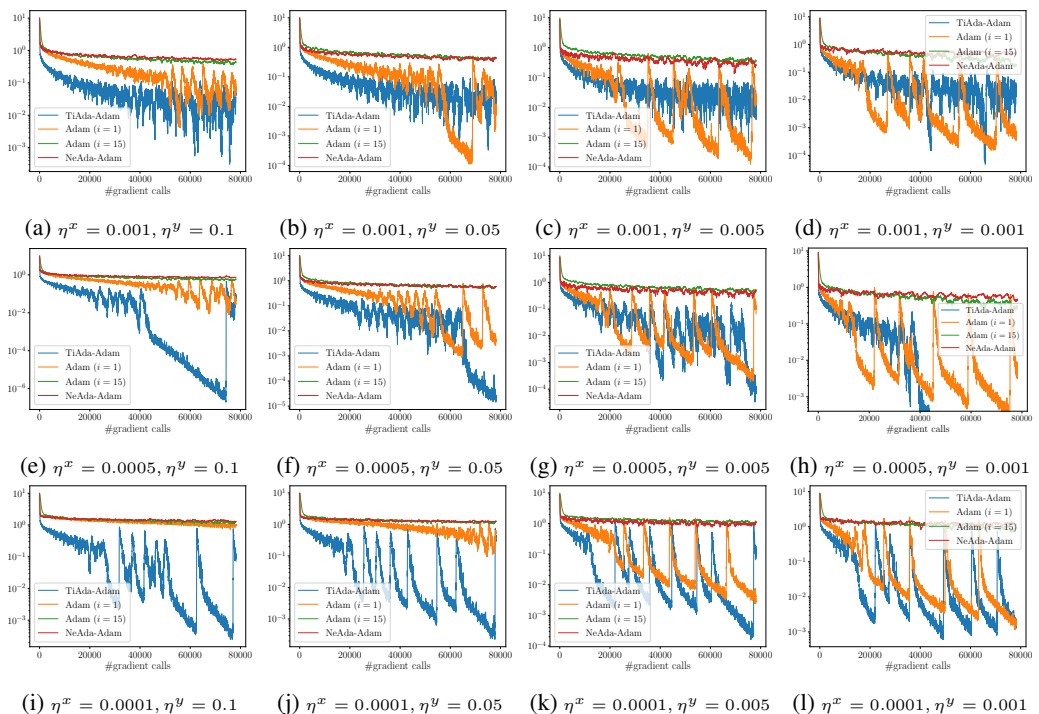

Figure 7: Gradient norms in $x$ of Adam-like algorithms on distributional robustness optimization (5). We use $i$ in the legend to indicate the number of inner loops.

## C.1 PROOF OF THEOREM 3.1

We present a formal version of Theorem 3.1.

**Theorem C.1** (deterministic setting). *Under Assumptions 3.1 to 3.3, Algorithm 1 with deterministic gradient oracles satisfies that for any $0 < \beta < \alpha < 1$, after $T$ iterations,*

$$\sum_{t=0}^{T-1} \|\nabla_x f(x_t, y_t)\|^2 \leq \max\{5C_1, 2C_2\},$$

*where*

$$C_1 = v_0^x + \left(\frac{2\Delta\Phi}{\eta^x}\right)^{\frac{1}{1-\alpha}} + \left(\frac{4\kappa l e^{(1-\alpha)(1-\log v_0^x)/2}}{e(1-\alpha)(v_0^x)^{2\alpha-1}}\right)^{\frac{2}{1-\alpha}} \mathbf{1}_{2\alpha\geq 1} + \left(\frac{2\kappa l}{1-2\alpha}\right)^{\frac{1}{\alpha}} \mathbf{1}_{2\alpha<1}$$

$$+ \left(\frac{c_1 c_5}{\eta^x}\right)^{\frac{1}{1-\alpha}} + \left(\frac{2c_1 c_4 \eta^x e^{(1-\alpha)(1-\log v_0^x)/2}}{e(1-\alpha)(v_0^x)^{2\alpha-\beta-1}}\right)^{\frac{2}{1-\alpha}} \mathbf{1}_{2\alpha-\beta\geq 1} + \left(\frac{c_1 c_4 \eta^x}{1-2\alpha+\beta}\right)^{\frac{1}{\alpha-\beta}} \mathbf{1}_{2\alpha-\beta<1}$$

$$C_2 = v_0^x + \left[\left(\frac{2\Delta\Phi + c_1 c_5}{\eta^x (v_0^x)^{1-2\alpha+\beta}} + \frac{c_1 c_4 \eta^x}{1-2\alpha+\beta} + \frac{2\kappa l e^{(1-2\alpha+\beta)(1-\log v_0^x)}}{e(1-2\alpha+\beta)(v_0^x)^{2\alpha-1}}\mathbf{1}_{2\alpha\geq 1} + \frac{2\kappa l}{(1-2\alpha)(v_0^x)^\beta}\mathbf{1}_{2\alpha<1}\right)\right.$$

$$\left(\frac{c_5}{(v_0^x)^{1-2\alpha+\beta}} + \frac{c_4 (\eta^x)^2}{1-2\alpha+\beta}\right)^{\frac{\alpha}{1-\beta}}\right]^{\frac{1}{1-(1-2\alpha+\beta)\left(1+\frac{\alpha}{1-\beta}\right)}} \mathbf{1}_{2\alpha-\beta<1}$$

$$+ \left[\left(\frac{2\Delta\Phi + c_1 c_5}{\eta^x (v_0^x)^{1/4}} + \frac{8\kappa l e^{(1-\log v_0^x)/4}}{e(v_0^x)^{2\alpha-1}} + \frac{4c_1 c_4 \eta^x e^{(1-\log v_0^x)/4}}{e(v_0^x)^{2\alpha-\beta-1}}\right)\right.$$

$$\left.\left(\frac{c_5}{(v_0^x)^{\frac{(1-\beta)}{4\alpha}}} + \frac{4c_4\alpha(\eta^x)^2 e^{(1-\beta)(1-\log v_0^x)/(4\alpha)}}{e(1-\beta)(v_0^x)^{2\alpha-\beta-1}}\right)^{\frac{\alpha}{1-\beta}}\right]^2 \mathbf{1}_{2\alpha\geq 1},$$

*with* $\Delta\Phi = \Phi(x_0) - \Phi^*$, $\quad c_1 = \dfrac{\eta^x \kappa^2}{\eta^y \left(v_{t_0}^y\right)^{\alpha-\beta}}$, $\quad c_2 = \max\left\{\dfrac{4\eta^y \mu l}{\mu + l}, \eta^y(\mu + l)\right\}$,

$$c_3 = 4(\mu + l)\left(\frac{1}{\mu^2} + \frac{\eta^y}{\left(v_{t_0}^y\right)^\beta}\right)c_2^{1/\beta}, \quad c_4 = (\mu + l)\left(\frac{2\kappa^2}{(v_0^y)^\alpha} + \frac{(\mu + l)\kappa^2}{\eta^y \mu l}\right),$$

$$c_5 = c_3 + \frac{\eta^y v_0^y}{(v_0^y)^\beta} + \frac{\eta^y c_2^{\frac{1-\beta}{\beta}}}{1 - \beta}.$$

*In addition, denoting the above upper bound for* $\sum_{t=0}^{T-1}\|\nabla_x f(x_t, y_t)\|^2$ *as* $C_3$, *we have*

$$\sum_{t=0}^{T-1}\|\nabla_y f(x_t, y_t)\|^2 \le \left(c_5 + c_4\left(\eta^x\right)^2\left(\frac{1 + \log C_3 - \log v_0^x}{(v_0^x)^{2\alpha-\beta-1}}\mathbf{1}_{2\alpha-\beta\ge1} + \frac{C_3^{1-2\alpha+\beta}}{1 - 2\alpha + \beta}\mathbf{1}_{2\alpha-\beta<1}\right)\right)^{\frac{1}{1-\beta}}.$$

*Proof.* Let us start from the smoothness of the primal function $\Phi(\cdot)$. By Lemma B.2,

$\Phi(x_{t+1})$

$\le \Phi(x_t) - \eta_t\langle\Phi(x_{t+1}), \nabla_x f(x_t, y_t)\rangle + kl\eta_t^2\|\nabla_x f(x_t, y_t)\|^2$

$= \Phi(x_t) - \eta_t\|\nabla_x f(x_t, y_t)\|^2 + \eta_t\langle\nabla_x f(x_t, y_t) - \nabla\Phi(x_t), \nabla_x f(x_t, y_t)\rangle + kl\eta_t^2\|\nabla_x f(x_t, y_t)\|^2$

$\le \Phi(x_t) - \eta_t\|\nabla_x f(x_t, y_t)\|^2 + \dfrac{\eta_t}{2}\|\nabla_x f(x_t, y_t)\|^2 + \dfrac{\eta_t}{2}\|\nabla_x f(x_t, y_t) - \nabla\Phi(x_t)\|^2 + kl\eta_t^2\|\nabla_x f(x_t, y_t)\|^2$

$= \Phi(x_t) - \dfrac{\eta_t}{2}\|\nabla_x f(x_t, y_t)\|^2 + kl\eta_t^2\|\nabla_x f(x_t, y_t)\|^2 + \dfrac{\eta_t}{2}\|\nabla_x f(x_t, y_t) - \nabla\Phi(x_t)\|^2$

$= \Phi(x_t) - \dfrac{\eta_t}{2}\|\nabla_x f(x_t, y_t)\|^2 + kl\eta_t^2\|\nabla_x f(x_t, y_t)\|^2 + \dfrac{\eta^x}{2\max\left\{v_{t+1}^x, v_{t+1}^y\right\}^\alpha}\|\nabla_x f(x_t, y_t) - \nabla\Phi(x_t)\|^2$

$\le \Phi(x_t) - \dfrac{\eta_t}{2}\|\nabla_x f(x_t, y_t)\|^2 + kl\eta_t^2\|\nabla_x f(x_t, y_t)\|^2 + \dfrac{\eta^x}{2\left(v_{t_0}^y\right)^{\alpha-\beta}\left(v_{t+1}^y\right)^\beta}\|\nabla_x f(x_t, y_t) - \nabla\Phi(x_t)\|^2$

$\le \Phi(x_t) - \dfrac{\eta_t}{2}\|\nabla_x f(x_t, y_t)\|^2 + kl\eta_t^2\|\nabla_x f(x_t, y_t)\|^2 + \dfrac{\eta^x \kappa^2}{2\left(v_{t_0}^y\right)^{\alpha-\beta}\left(v_{t+1}^y\right)^\beta}\|\nabla_y f(x_t, y_t)\|^2$

$\le \Phi(x_t) - \dfrac{\eta_t}{2}\|\nabla_x f(x_t, y_t)\|^2 + kl\eta_t^2\|\nabla_x f(x_t, y_t)\|^2 + \dfrac{\eta^x \kappa^2}{2\eta^y\left(v_{t_0}^y\right)^{\alpha-\beta}}\cdot\gamma_t\|\nabla_y f(x_t, y_t)\|^2,$

where in the second to last inequality, we used the strong-concavity of $f(x, \cdot)$:

$$\|\nabla_x f(x_t, y_t) - \nabla\Phi(x_t)\| \le l\|y_t - y_t^*\| \le \kappa\|\nabla_y f(x_t, y_t)\|.$$

Telescoping and rearranging the terms, we have

$$\sum_{t=0}^{T-1}\eta_t\|\nabla_x f(x_t, y_t)\|^2$$

$$\le 2\underbrace{(\Phi(x_0) - \Phi^*)}_{\Delta\Phi} + 2\kappa l\sum_{t=0}^{T-1}\eta_t^2\|\nabla_x f(x_t, y_t)\|^2 + \underbrace{\frac{\eta^x \kappa^2}{\eta^y\left(v_{t_0}^y\right)^{\alpha-\beta}}}_{c_1}\sum_{t=0}^{T-1}\gamma_t\|\nabla_y f(x_t, y_t)\|^2$$

$$= 2\Delta\Phi + \sum_{t=0}^{T-1}\frac{2\kappa l\eta^x}{\max\left\{v_{t+1}^x, v_{t+1}^y\right\}^{2\alpha}}\|\nabla_x f(x_t, y_t)\|^2 + c_1\sum_{t=0}^{T-1}\gamma_t\|\nabla_y f(x_t, y_t)\|^2$$

$$\le 2\Delta\Phi + \sum_{t=0}^{T-1}\frac{2\kappa l\eta^x}{\left(v_{t+1}^x\right)^{2\alpha}}\|\nabla_x f(x_t, y_t)\|^2 + c_1\sum_{t=0}^{T-1}\gamma_t\|\nabla_y f(x_t, y_t)\|^2$$

$$\le 2\Delta\Phi + 2\kappa l\eta^x\left(\frac{1 + \log v_T^x - \log v_0^x}{(v_0^x)^{2\alpha-1}}\cdot\mathbf{1}_{2\alpha\ge1} + \frac{(v_T^x)^{1-2\alpha}}{1 - 2\alpha}\cdot\mathbf{1}_{2\alpha<1}\right) + c_1\sum_{t=0}^{T-1}\gamma_t\|\nabla_y f(x_t, y_t)\|^2.$$

$$\tag{6}$$

We proceed to bound $\sum_{t=0}^{T-1} \gamma_t \|\nabla_y f(x_t, y_t)\|^2$. Let $t_0$ be the first iteration such that $\left(v_{t_0+1}^y\right)^\beta > c_2 := \max\left\{\frac{4\eta^y \mu l}{\mu+l}, \eta^y(\mu+l)\right\}$. We have $v_{t_0}^y \le c_2^{1/\beta}$, and for $t \ge t_0$,

$$
\left\|y_{t+1} - y_{t+1}^*\right\|^2
$$

$$
\le (1+\lambda_t)\|y_{t+1} - y_t^*\|^2 + \left(1 + \frac{1}{\lambda_t}\right)\left\|y_{t+1}^* - y_t^*\right\|^2
$$

$$
\le \underbrace{(1+\lambda_t)\left(\|y_t - y_t^*\|^2 + \frac{(\eta^y)^2}{\left(v_{t+1}^y\right)^{2\beta}}\|\nabla_y f(x_t, y_t)\|^2 + \frac{2\eta^y}{\left(v_{t+1}^y\right)^\beta}\langle y_t - y_t^*, \nabla_y f(x_t, y_t)\rangle\right)}_{(A)}
$$

$$
+ \left(1 + \frac{1}{\lambda_t}\right)\left\|y_{t+1}^* - y_t^*\right\|^2,
$$

where $\lambda_t > 0$ will be determined later. For $l$-smooth and $\mu$-strongly convex function $g(x)$, according to Theorem 2.1.12 in Nesterov (2003), we have

$$
\langle \nabla g(x) - \nabla g(y), x - y\rangle \ge \frac{\mu l}{\mu+l}\|x-y\|^2 + \frac{1}{\mu+l}\|\nabla g(x) - \nabla g(y)\|^2.
$$

Therefore,

Term (A)

$$
\le (1+\lambda_t)\left(\left(1 - \frac{2\eta^y \mu l}{(\mu+l)\left(v_{t+1}^y\right)^\beta}\right)\|y_t - y_t^*\|^2 + \left(\frac{(\eta^y)^2}{\left(v_{t+1}^y\right)^{2\beta}} - \frac{2\eta^y}{(\mu+l)\left(v_{t+1}^y\right)^\beta}\right)\|\nabla_y f(x_t, y_t)\|^2\right).
$$

Let $\lambda_t = \frac{\eta^y \mu l}{(\mu+l)\left(v_{t+1}^y\right)^\beta - 2\eta^y \mu l}$. Note that $\lambda_t > 0$ after $t_0$. Then

Term (A)

$$
\le \left(1 - \frac{\eta^y \mu l}{(\mu+l)\left(v_{t+1}^y\right)^\beta}\right)\|y_t - y_t^*\|^2 + (1+\lambda_t)\left(\frac{(\eta^y)^2}{\left(v_{t+1}^y\right)^{2\beta}} - \frac{2\eta^y}{(\mu+l)\left(v_{t+1}^y\right)^\beta}\right)\|\nabla_y f(x_t, y_t)\|^2
$$

$$
\le \|y_t - y_t^*\|^2 + \underbrace{(1+\lambda_t)\left(\frac{(\eta^y)^2}{\left(v_{t+1}^y\right)^{2\beta}} - \frac{2\eta^y}{(\mu+l)\left(v_{t+1}^y\right)^\beta}\right)}_{(B)}\|\nabla_y f(x_t, y_t)\|^2.
$$

As $1 + \lambda_t \ge 1$ and $\left(v_{t+1}^y\right)^\beta \ge \eta^y(\mu+l)$, we have term (B) $\le -\frac{\eta^y}{(\mu+l)\left(v_{t+1}^y\right)^\beta}$. Putting them back, we can get

$$
\left\|y_{t+1} - y_{t+1}^*\right\|^2
$$

$$
\le \|y_t - y_t^*\|^2 - \frac{\eta^y}{(\mu+l)\left(v_{t+1}^y\right)^\beta}\|\nabla_y f(x_t, y_t)\|^2 + \left(1 + \frac{1}{\lambda_t}\right)\left\|y_{t+1}^* - y_t^*\right\|^2
$$

$$
\le \|y_t - y_t^*\|^2 - \frac{\eta^y}{(\mu+l)\left(v_{t+1}^y\right)^\beta}\|\nabla_y f(x_t, y_t)\|^2 + \frac{(\mu+l)\left(v_{t+1}^y\right)^\beta}{\eta^y \mu l}\left\|y_{t+1}^* - y_t^*\right\|^2
$$

$$
\le \|y_t - y_t^*\|^2 - \frac{\eta^y}{(\mu+l)\left(v_{t+1}^y\right)^\beta}\|\nabla_y f(x_t, y_t)\|^2 + \frac{(\mu+l)\kappa^2 \left(v_{t+1}^y\right)^\beta}{\eta^y \mu l}\|x_{x+1} - x_t\|^2
$$

$$
= \|y_t - y_t^*\|^2 - \frac{\eta^y}{(\mu+l)\left(v_{t+1}^y\right)^\beta}\|\nabla_y f(x_t, y_t)\|^2 + \frac{(\mu+l)\kappa^2 \left(v_{t+1}^y\right)^\beta \eta_t^2}{\eta^y \mu l}\|\nabla_x f(x_t, y_t)\|^2.
$$

Then, by telescoping, we have

$$\sum_{t=t_0}^{T-1} \frac{\eta^y}{(\mu+l)\left(v_{t+1}^y\right)^{\beta}} \|\nabla_y f(x_t, y_t)\|^2 \leq \|y_{t_0} - y_{t_0}^*\|^2 + \sum_{t=t_0}^{T-1} \frac{(\mu+l)\kappa^2 \left(v_{t+1}^y\right)^{\beta} \eta_t^2}{\eta^y \mu l} \|\nabla_x f(x_t, y_t)\|^2. \tag{7}$$

For the first term in the RHS, using Young's inequality with $\tau$ to be determined later, we have

$$\|y_{t_0} - y_{t_0}^*\|^2 \leq 2\|y_{t_0} - y_{t_0-1}^*\|^2 + 2\|y_{t_0}^* - y_{t_0-1}^*\|^2$$

$$= 2\left\|\mathcal{P}_{\mathcal{Y}}\left(y_{t_0-1} + \gamma_{t_0-1}\nabla_y f(x_{t_0-1}, y_{t_0-1})\right) - y_{t_0-1}^*\right\|^2 + 2\|y_{t_0}^* - y_{t_0-1}^*\|^2$$

$$\leq 2\left\|y_{t_0-1} + \gamma_{t_0-1}\nabla_y f(x_{t_0-1}, y_{t_0-1}) - y_{t_0-1}^*\right\|^2 + 2\|y_{t_0}^* - y_{t_0-1}^*\|^2$$

$$\leq 4\left(\|y_{t_0-1} - y_{t_0-1}^*\|^2 + \gamma_{t_0-1}^2\|\nabla_y f(x_{t_0-1}, y_{t_0-1})\|^2\right) + 2\|y_{t_0}^* - y_{t_0-1}^*\|^2$$

$$\leq 4\left(\frac{1}{\mu^2}\|\nabla_y f(x_{t_0-1}, y_{t_0-1})\|^2 + \gamma_{t_0-1}^2\|\nabla_y f(x_{t_0-1}, y_{t_0-1})\|^2\right) + 2\|y_{t_0}^* - y_{t_0-1}^*\|^2$$

$$= 4\left(\frac{1}{\mu^2} + \gamma_{t_0-1}^2\right)\|\nabla_y f(x_{t_0-1}, y_{t_0-1})\|^2 + 2\|y_{t_0}^* - y_{t_0-1}^*\|^2$$

$$\leq 4\left(\frac{1}{\mu^2} + \gamma_0^2\right)v_{t_0}^y + 2\|y_{t_0}^* - y_{t_0-1}^*\|^2$$

$$\leq 4\left(\frac{1}{\mu^2} + \frac{\eta^y}{\left(v_{t_0}^y\right)^{\beta}}\right)c_2^{1/\beta} + 2\|y_{t_0}^* - y_{t_0-1}^*\|^2$$

$$\leq 4\left(\frac{1}{\mu^2} + \frac{\eta^y}{\left(v_{t_0}^y\right)^{\beta}}\right)c_2^{1/\beta} + 2\kappa^2\|x_{t_0} - x_{t_0-1}\|^2$$

$$\leq 4\left(\frac{1}{\mu^2} + \frac{\eta^y}{\left(v_{t_0}^y\right)^{\beta}}\right)c_2^{1/\beta} + 2\kappa^2\eta_{t_0-1}^2\|\nabla_x f(x_{t_0-1}, y_{t_0-1})\|^2$$

$$\leq 4\left(\frac{1}{\mu^2} + \frac{\eta^y}{\left(v_{t_0}^y\right)^{\beta}}\right)c_2^{1/\beta} + \frac{2\kappa^2\left(v_{t+1}^y\right)^{\beta}}{\left(v_0^y\right)^{\beta}}\eta_{t_0-1}^2\|\nabla_x f(x_{t_0-1}, y_{t_0-1})\|^2.$$

Combined with Equation (7), we have

$$\sum_{t=t_0}^{T-1} \frac{\eta^y}{\left(v_{t+1}^y\right)^{\beta}}\|\nabla_y f(x_t, y_t)\|^2$$

$$\leq \underbrace{4(\mu+l)\left(\frac{1}{\mu^2} + \frac{\eta^y}{\left(v_{t_0}^y\right)^{\beta}}\right)c_2^{1/\beta}}_{c_3} + \underbrace{(\mu+l)\left(\frac{2\kappa^2}{(v_0^y)^{\alpha}} + \frac{(\mu+l)\kappa^2}{\eta^y\mu l}\right)}_{c_4}\sum_{t=t_0-1}^{T-1}\left(v_{t+1}^y\right)^{\beta}\eta_t^2\|\nabla_x f(x_t, y_t)\|^2.$$

By adding terms from 0 to $t_0 - 1$ and $\frac{\eta^y v_0^y}{\left(v_0^y\right)^{\beta}}$ from both sides, we have

$$\frac{\eta^y v_0^y}{\left(v_0^y\right)^{\beta}} + \sum_{t=0}^{T-1}\frac{\eta^y}{\left(v_{t+1}^y\right)^{\beta}}\|\nabla_y f(x_t, y_t)\|^2$$

$$\leq c_3 + \frac{\eta^y v_0^y}{\left(v_0^y\right)^{\beta}} + c_4\sum_{t=0}^{T-1}\left(v_{t+1}^y\right)^{\beta}\eta_t^2\|\nabla_x f(x_t, y_t)\|^2 + \sum_{t=t=0}^{t_0-1}\frac{\eta^y}{\left(v_{t+1}^y\right)^{\beta}}\|\nabla_y f(x_t, y_t)\|^2$$

$$\leq c_3 + \frac{\eta^y v_0^y}{\left(v_0^y\right)^{\beta}} + c_4\sum_{t=0}^{T-1}\left(v_{t+1}^y\right)^{\beta}\eta_t^2\|\nabla_x f(x_t, y_t)\|^2 + \frac{\eta^y v_0^y}{\left(v_0^y\right)^{\beta}} + \sum_{t=t=0}^{t_0-1}\frac{\eta^y}{\left(v_{t+1}^y\right)^{\beta}}\|\nabla_y f(x_t, y_t)\|^2$$

$$\leq c_3 + \frac{\eta^y v_0^y}{\left(v_0^y\right)^{\beta}} + c_4\sum_{t=0}^{T-1}\left(v_{t+1}^y\right)^{\beta}\eta_t^2\|\nabla_x f(x_t, y_t)\|^2 + \frac{\eta^y}{1-\beta}v_{t_0}^{1-\beta}$$

$$\leq c_3 + \frac{\eta^y v_0^y}{(v_0^y)^\beta} + c_4 \sum_{t=0}^{T-1} \left(v_{t+1}^y\right)^\beta \eta_t^2 \|\nabla_x f(x_t, y_t)\|^2 + \frac{\eta^y c_2^{\frac{1-\beta}{\beta}}}{1-\beta}$$

$$= c_3 + \frac{\eta^y v_0^y}{(v_0^y)^\beta} + \frac{\eta^y c_2^{\frac{1-\beta}{\beta}}}{1-\beta} + c_4 \left(\eta^x\right)^2 \sum_{t=0}^{T-1} \frac{\left(v_{t+1}^y\right)^\beta}{\max\left\{v_{t+1}^x, v_{t+1}^y\right\}^{2\alpha}} \|\nabla_x f(x_t, y_t)\|^2$$

$$= \underbrace{c_3 + \frac{\eta^y v_0^y}{(v_0^y)^\beta} + \frac{\eta^y c_2^{\frac{1-\beta}{\beta}}}{1-\beta}}_{c_5} + c_4 \left(\eta^x\right)^2 \sum_{t=0}^{T-1} \frac{1}{\left(v_{t+1}^x\right)^{2\alpha-\beta}} \|\nabla_x f(x_t, y_t)\|^2$$

$$\leq c_5 + c_4 \left(\eta^x\right)^2 \left(\frac{1 + \log v_T^x - \log v_0^x}{(v_0^x)^{2\alpha-\beta-1}} \cdot \mathbf{1}_{2\alpha-\beta\geq 1} + \frac{(v_T^x)^{1-2\alpha+\beta}}{1-2\alpha+\beta} \cdot \mathbf{1}_{2\alpha-\beta<1}\right).$$

The LHS can be bounded by $(v_T^y)^{1-\beta}$ by Lemma B.1. Then we get two useful inequalities from above:

$$\begin{cases} \sum_{t=0}^{T-1} \gamma_t \|\nabla_y f(x_t, y_t)\|^2 \leq c_5 + c_4 \left(\eta^x\right)^2 \left(\frac{1+\log v_T^x - \log v_0^x}{(v_0^x)^{2\alpha-\beta-1}} \cdot \mathbf{1}_{2\alpha-\beta\geq 1} + \frac{(v_T^x)^{1-2\alpha+\beta}}{1-2\alpha+\beta} \cdot \mathbf{1}_{2\alpha-\beta<1}\right) \\ v_T^y \leq \left(c_5 + c_4 \left(\eta^x\right)^2 \left(\frac{1+\log v_T^x - \log v_0^x}{(v_0^x)^{2\alpha-\beta-1}} \cdot \mathbf{1}_{2\alpha-\beta\geq 1} + \frac{(v_T^x)^{1-2\alpha+\beta}}{1-2\alpha+\beta} \cdot \mathbf{1}_{2\alpha-\beta<1}\right)\right)^{\frac{1}{1-\beta}}. \end{cases}$$

$$(8)$$

Now bring it back to Equation (6), we get

$$\sum_{t=0}^{T-1} \eta_t \|\nabla_x f(x_t, y_t)\|^2$$

$$\leq 2\Delta\Phi + 2\kappa l \eta^x \left(\frac{1 + \log v_T^x - \log v_0^x}{(v_0^x)^{2\alpha-1}} \cdot \mathbf{1}_{2\alpha\geq 1} + \frac{(v_T^x)^{1-2\alpha}}{1-2\alpha} \cdot \mathbf{1}_{2\alpha<1}\right)$$

$$+ c_1 c_5 + c_1 c_4 \left(\eta^x\right)^2 \left(\frac{1 + \log v_T^x - \log v_0^x}{(v_0^x)^{2\alpha-\beta-1}} \cdot \mathbf{1}_{2\alpha-\beta\geq 1} + \frac{(v_T^x)^{1-2\alpha+\beta}}{1-2\alpha+\beta} \cdot \mathbf{1}_{2\alpha-\beta<1}\right).$$

For the LHS, we have

$$\sum_{t=0}^{T-1} \eta_t \|\nabla_x f(x_t, y_t)\|^2 = \sum_{t=0}^{T-1} \frac{\eta^x}{\max\left\{v_{t+1}^x, v_{t+1}^y\right\}^\alpha} \|\nabla_x f(x_t, y_t)\|^2$$

$$\geq \frac{\eta^x}{\max\left\{v_T^x, v_T^y\right\}^\alpha} \sum_{t=0}^{T-1} \|\nabla_x f(x_t, y_t)\|^2$$

From here, by combining two inequalites above and noting that $\sum_{t=0}^{T-1} \|\nabla_x f(x_t, y_t)\|^2 \leq v_T^x$, we can already conclude that $\sum_{t=0}^{T-1} \|\nabla_x f(x_t, y_t)\|^2 = \mathcal{O}(1)$. Now we will provide an explicit bound. We consider two cases:

**(1)** If $v_T^y \leq v_T^x$, then

$$\sum_{t=0}^{T-1} \|\nabla_x f(x_t, y_t)\|^2$$

$$\leq \frac{2\Delta\Phi (v_T^x)^\alpha}{\eta^x} + 2\kappa l \left(\frac{(v_T^x)^\alpha (1 + \log v_T^x - \log v_0^x)}{(v_0^x)^{2\alpha-1}} \cdot \mathbf{1}_{2\alpha\geq 1} + \frac{(v_T^x)^{1-\alpha}}{1-2\alpha} \cdot \mathbf{1}_{2\alpha<1}\right)$$

$$+ \frac{c_1 c_5 (v_T^x)^\alpha}{\eta^x} + c_1 c_4 \eta^x \left(\frac{(v_T^x)^\alpha (1 + \log v_T^x - \log v_0^x)}{(v_0^x)^{2\alpha-\beta-1}} \cdot \mathbf{1}_{2\alpha-\beta\geq 1} + \frac{(v_T^x)^{1-\alpha+\beta}}{1-2\alpha+\beta} \cdot \mathbf{1}_{2\alpha-\beta<1}\right)$$

$$
= \frac{2\Delta\Phi\,(v_T^x)^\alpha}{\eta^x} + 2\kappa l \left( \frac{(v_T^x)^\alpha\,(v_T^x)^{\frac{1-\alpha}{2}}\,(v_T^x)^{\frac{\alpha-1}{2}}\,(1+\log v_T^x - \log v_0^x)}{(v_0^x)^{2\alpha-1}} \cdot \mathbf{1}_{2\alpha\geq1} + \frac{(v_T^x)^{1-\alpha}}{1-2\alpha} \cdot \mathbf{1}_{2\alpha<1} \right)
$$

$$
+ \frac{c_1 c_5\,(v_T^x)^\alpha}{\eta^x} + c_1 c_4 \eta^x \left( \frac{(v_T^x)^\alpha\,(v_T^x)^{\frac{1-\alpha}{2}}\,(v_T^x)^{\frac{\alpha-1}{2}}\,(1+\log v_T^x - \log v_0^x)}{(v_0^x)^{2\alpha-\beta-1}} \cdot \mathbf{1}_{2\alpha-\beta\geq1} + \frac{(v_T^x)^{1-\alpha+\beta}}{1-2\alpha+\beta} \cdot \mathbf{1}_{2\alpha-\beta<1} \right)
$$

$$
\leq \frac{2\Delta\Phi\,(v_T^x)^\alpha}{\eta^x} + 2\kappa l \left( \frac{2e^{(1-\alpha)(1-\log v_0^x)/2}\,(v_T^x)^{\frac{1+\alpha}{2}}}{e(1-\alpha)\,(v_0^x)^{2\alpha-1}} \cdot \mathbf{1}_{2\alpha\geq1} + \frac{(v_T^x)^{1-\alpha}}{1-2\alpha} \cdot \mathbf{1}_{2\alpha<1} \right)
$$

$$
+ \frac{c_1 c_5\,(v_T^x)^\alpha}{\eta^x} + c_1 c_4 \eta^x \left( \frac{2e^{(1-\alpha)(1-\log v_0^x)/2}\,(v_T^x)^{\frac{1+\alpha}{2}}}{e(1-\alpha)\,(v_0^x)^{2\alpha-\beta-1}} \cdot \mathbf{1}_{2\alpha-\beta\geq1} + \frac{(v_T^x)^{1-\alpha+\beta}}{1-2\alpha+\beta} \cdot \mathbf{1}_{2\alpha-\beta<1} \right),
$$

$$(9)$$

where we used $x^{-m}(c + \log x) \leq \frac{e^{cm}}{em}$ for $x > 0$, $m > 0$ and $c \in \mathbb{R}$ in the last inequality. Also, if $0 < \alpha_i < 1$ and $b_i$ are positive constants, and $x \leq \sum_{i=1}^n b_i x^{\alpha_i}$, then we get $x \leq n \sum_{i=1}^n b_i^{1/(1-\alpha_i)}$. Now consider $v_T^x$ as the $x$ in the previous statement, and note that the LHS of Equation (9) equals to $v_T^x - v_0^x$. Then we can get

$$
v_T^x \leq 5v_0^x + 5\left(\frac{2\Delta\Phi}{\eta^x}\right)^{\frac{1}{1-\alpha}} + 5\left(\frac{4\kappa l e^{(1-\alpha)(1-\log v_0^x)/2}}{e(1-\alpha)\,(v_0^x)^{2\alpha-1}}\right)^{\frac{2}{1-\alpha}} \cdot \mathbf{1}_{2\alpha\geq1} + 5\left(\frac{2\kappa l}{1-2\alpha}\right)^{\frac{1}{\alpha}} \cdot \mathbf{1}_{2\alpha<1}
$$

$$
+ 5\left(\frac{c_1 c_5}{\eta^x}\right)^{\frac{1}{1-\alpha}} + 5\left(\frac{2c_1 c_4 \eta^x e^{(1-\alpha)(1-\log v_0^x)/2}}{e(1-\alpha)\,(v_0^x)^{2\alpha-\beta-1}}\right)^{\frac{2}{1-\alpha}} \cdot \mathbf{1}_{2\alpha-\beta\geq1} + 5\left(\frac{c_1 c_4 \eta^x}{1-2\alpha+\beta}\right)^{\frac{1}{\alpha-\beta}} \cdot \mathbf{1}_{2\alpha-\beta<1}.
$$

$$(10)$$

Note that the RHS is a constant and also an upper bound for $\sum_{t=0}^{T-1}\|\nabla_x f(x_t, y_t)\|^2$.

**(2)** If $v_T^y \leq v_T^x$, then we can use the upper bound for $v_T^y$ from Equation (8). We now discuss two cases:

1. $2\alpha < 1 + \beta$. Then we have

$$
\sum_{t=0}^{T-1}\|\nabla_x f(x_t, y_t)\|^2
$$

$$
\leq \left(\frac{2\Delta\Phi + c_1 c_5}{\eta^x} + 2\kappa l\left(\frac{1+\log v_T^x - \log v_0^x}{(v_0^x)^{2\alpha-1}} \cdot \mathbf{1}_{2\alpha\geq1} + \frac{(v_T^x)^{1-2\alpha}}{1-2\alpha} \cdot \mathbf{1}_{2\alpha<1}\right) + \frac{c_1 c_4 \eta^x\,(v_T^x)^{1-2\alpha+\beta}}{1-2\alpha+\beta}\right)
$$

$$
\left(c_5 + \frac{c_4\,(\eta^x)^2\,(v_T^x)^{1-2\alpha+\beta}}{1-2\alpha+\beta}\right)^{\frac{\alpha}{1-\beta}}
$$

$$
\leq \left(\frac{2\Delta\Phi + c_1 c_5}{\eta^x\,(v_0^x)^{1-2\alpha+\beta}} + 2\kappa l\left(\frac{1+\log v_T^x - \log v_0^x}{(v_0^x)^{2\alpha-1}\,(v_T^x)^{1-2\alpha+\beta}} \cdot \mathbf{1}_{2\alpha\geq1} + \frac{1}{(1-2\alpha)\,(v_0^x)^\beta} \cdot \mathbf{1}_{2\alpha<1}\right) + \frac{c_1 c_4 \eta^x}{1-2\alpha+\beta}\right)
$$

$$
\left(\frac{c_5}{(v_0^x)^{1-2\alpha+\beta}} + \frac{c_4\,(\eta^x)^2}{1-2\alpha+\beta}\right)^{\frac{\alpha}{1-\beta}} \cdot (v_T^x)^{1-2\alpha+\beta+\frac{(1-2\alpha+\beta)\alpha}{1-\beta}}
$$

$$
\leq \left(\frac{2\Delta\Phi + c_1 c_5}{\eta^x\,(v_0^x)^{1-2\alpha+\beta}} + 2\kappa l\left(\frac{e^{(1-2\alpha+\beta)(1-\log v_0^x)}}{e(1-2\alpha+\beta)\,(v_0^x)^{2\alpha-1}} \cdot \mathbf{1}_{2\alpha\geq1} + \frac{1}{(1-2\alpha)\,(v_0^x)^\beta} \cdot \mathbf{1}_{2\alpha<1}\right) + \frac{c_1 c_4 \eta^x}{1-2\alpha+\beta}\right)
$$

$$
\left(\frac{c_5}{(v_0^x)^{1-2\alpha+\beta}} + \frac{c_4\,(\eta^x)^2}{1-2\alpha+\beta}\right)^{\frac{\alpha}{1-\beta}} \cdot (v_T^x)^{1-2\alpha+\beta+\frac{(1-2\alpha+\beta)\alpha}{1-\beta}},
$$

Note that since $\alpha > \beta$, we have

$$
1 - 2\alpha + \beta + \frac{(1-2\alpha+\beta)\alpha}{1-\beta} \leq \frac{(1-\alpha)\alpha}{1-\beta} + 1 - \alpha = 1 + \frac{\alpha(\beta-\alpha)}{1-\beta} < 1.
$$

Therefore, with the same reasoning as Equation (10),

$$\sum_{t=0}^{T-1}\|\nabla_x f(x_t,y_t)\|^2 \le v_T^x$$

$$\le 2\Bigg[\left(\frac{2\Delta\Phi + c_1 c_5}{\eta^x\,(v_0^x)^{1-2\alpha+\beta}} + \frac{c_1 c_4 \eta^x}{1-2\alpha+\beta} + \frac{2\kappa l e^{(1-2\alpha+\beta)(1-\log v_0^x)}}{e(1-2\alpha+\beta)\,(v_0^x)^{2\alpha-1}}\cdot \mathbf{1}_{2\alpha\ge 1} + \frac{2\kappa l}{(1-2\alpha)\,(v_0^x)^{\beta}}\cdot \mathbf{1}_{2\alpha<1}\right)$$

$$\left(\frac{c_5}{(v_0^x)^{1-2\alpha+\beta}} + \frac{c_4\,(\eta^x)^2}{1-2\alpha+\beta}\right)^{\frac{\alpha}{1-\beta}}\,\Bigg]^{\frac{1}{1-(1-2\alpha+\beta)\left(1+\frac{\alpha}{1-\beta}\right)}} + 2v_0^x,$$

which gives us constant RHS.

2. $2\alpha \ge 1+\beta$. Then we have

$$\sum_{t=0}^{T-1}\|\nabla_x f(x_t,y_t)\|^2$$

$$\le \left(\frac{2\Delta\Phi + c_1 c_5}{\eta^x} + \frac{2\kappa l\,(1+\log v_T^x - \log v_0^x)}{(v_0^x)^{2\alpha-1}} + \frac{c_1 c_4 \eta^x\,(1+\log v_T^x - \log v_0^x)}{(v_0^x)^{2\alpha-\beta-1}}\right)$$

$$\left(c_5 + \frac{c_4\,(\eta^x)^2\,(1+\log v_T^x - \log v_0^x)}{(v_0^x)^{2\alpha-\beta-1}}\right)^{\frac{\alpha}{1-\beta}}$$

$$\le \left(\frac{2\Delta\Phi + c_1 c_5}{\eta^x\,(v_0^x)^{1/4}} + \frac{2\kappa l\,(1+\log v_T^x - \log v_0^x)}{(v_0^x)^{2\alpha-1}\,(v_T^x)^{1/4}} + \frac{c_1 c_4 \eta^x\,(1+\log v_T^x - \log v_0^x)}{(v_0^x)^{2\alpha-\beta-1}\,(v_T^x)^{1/4}}\right)$$

$$\left(\frac{c_5}{(v_0^x)^{\frac{(1-\beta)}{4\alpha}}} + \frac{c_4\,(\eta^x)^2\,(1+\log v_T^x - \log v_0^x)}{(v_0^x)^{2\alpha-\beta-1}\,(v_T^x)^{\frac{(1-\beta)}{4\alpha}}}\right)^{\frac{\alpha}{1-\beta}}\cdot (v_T^x)^{1/2}$$

$$\le \left(\frac{2\Delta\Phi + c_1 c_5}{\eta^x\,(v_0^x)^{1/4}} + \frac{8\kappa l e^{(1-\log v_0^x)/4}}{e\,(v_0^x)^{2\alpha-1}} + \frac{4 c_1 c_4 \eta^x e^{(1-\log v_0^x)/4}}{e\,(v_0^x)^{2\alpha-\beta-1}}\right)$$

$$\left(\frac{c_5}{(v_0^x)^{\frac{(1-\beta)}{4\alpha}}} + \frac{4 c_4 \alpha\,(\eta^x)^2\,e^{(1-\beta)(1-\log v_0^x)/(4\alpha)}}{e(1-\beta)\,(v_0^x)^{2\alpha-\beta-1}}\right)^{\frac{\alpha}{1-\beta}}\cdot (v_T^x)^{1/2},$$

which implies

$$\sum_{t=0}^{T-1}\|\nabla_x f(x_t,y_t)\|^2 \le v_T^x$$

$$\le 2\Bigg[\left(\frac{2\Delta\Phi + c_1 c_5}{\eta^x\,(v_0^x)^{1/4}} + \frac{8\kappa l e^{(1-\log v_0^x)/4}}{e\,(v_0^x)^{2\alpha-1}} + \frac{4 c_1 c_4 \eta^x e^{(1-\log v_0^x)/4}}{e\,(v_0^x)^{2\alpha-\beta-1}}\right)$$

$$\left(\frac{c_5}{(v_0^x)^{\frac{(1-\beta)}{4\alpha}}} + \frac{4 c_4 \alpha\,(\eta^x)^2\,e^{(1-\beta)(1-\log v_0^x)/(4\alpha)}}{e(1-\beta)\,(v_0^x)^{2\alpha-\beta-1}}\right)^{\frac{\alpha}{1-\beta}}\Bigg]^2 + 2v_0^x.$$

Now we also get only a constant on the RHS.

Summarizing all the cases, we finish the proof.

$\square$

## C.2 INTERMEDIATE LEMMAS FOR THEOREM 3.2

**Lemma C.1.** *Under the same setting as Theorem 3.2, if for $t = t_0$ to $t_1 - 1$ and any $\lambda_t > 0$, $S_t$,*

$$\left\|y_{t+1} - y_{t+1}^*\right\|^2 \le (1+\lambda_t)\left\|y_{t+1} - y_t^*\right\|^2 + S_t,$$

*then we have*

$$\mathbb{E}\left[\sum_{t=t_0}^{t_1-1}\left(f(x_t,y_t^*)-f(x_t,y_t)\right)\right]\leq\mathbb{E}\left[\sum_{t=t_0+1}^{t_1-1}\left(\frac{1-\gamma_t\mu}{2\gamma_t}\|y_t-y_t^*\|^2-\frac{1}{2\gamma_t(1+\lambda_t)}\left\|y_{t+1}-y_{t+1}^*\right\|^2\right)\right]$$

$$+\mathbb{E}\left[\sum_{t=t_0}^{t_1-1}\frac{\gamma_t}{2}\left\|\nabla_y\widetilde{f}(x_t,y_t)\right\|^2\right]+\mathbb{E}\left[\sum_{t=t_0}^{t_1-1}\frac{S_t}{2\gamma_t(1+\lambda_t)}\right].$$

*Proof.* Letting $\lambda_t:=\frac{\mu\eta^y}{2\left(v_{t+1}^y\right)^\beta}$, we have

$$\left\|y_{t+1}-y_{t+1}^*\right\|^2$$

$$\leq(1+\lambda_t)\|y_{t+1}-y_t^*\|^2+S_t$$

$$=(1+\lambda_t)\left\|\mathcal{P}_\mathcal{Y}\left(y_t+\gamma_t\nabla_y\widetilde{f}(x_t,y_t)\right)-y_t^*\right\|^2+S_t$$

$$\leq(1+\lambda_t)\left\|y_t+\gamma_t\nabla_y\widetilde{f}(x_t,y_t)-y_t^*\right\|^2+S_t$$

$$=(1+\lambda_t)\left(\|y_t-y_t^*\|^2+\gamma_t^2\left\|\nabla_y\widetilde{f}(x_t,y_t)\right\|^2+2\gamma_t\left\langle\nabla_y\widetilde{f}(x_t,y_t),y_t-y_t^*\right\rangle\right)+S_t$$

$$=(1+\lambda_t)\left(\|y_t-y_t^*\|^2+\gamma_t^2\left\|\nabla_y\widetilde{f}(x_t,y_t)\right\|^2+2\gamma_t\left\langle\nabla_y\widetilde{f}(x_t,y_t),y_t-y_t^*\right\rangle\right.$$

$$\left.+\gamma_t\mu\|y_t-y_t^*\|^2-\gamma_t\mu\|y_t-y_t^*\|^2\right)+S_t$$

By multiplying $\frac{1}{\gamma_t(1+\lambda_t)}$ and rearranging the terms, we can get

$$2\left\langle\nabla_y\widetilde{f}(x_t,y_t),y_t^*-y_t\right\rangle-\mu\|y_t-y_t^*\|^2$$

$$\leq\frac{1-\gamma_t\mu}{\gamma_t}\|y_t-y_t^*\|^2-\frac{1}{\gamma_t(1+\lambda_t)}\left\|y_{t+1}-y_{t+1}^*\right\|^2+\gamma_t\left\|\nabla_y\widetilde{f}(x_t,y_t)\right\|^2+\frac{S_t}{\gamma_t(1+\lambda_t)}.$$

By telescoping from $t=t_0$ to $t_1-1$, we have

$$\sum_{t=t_0}^{t_1-1}\left(\left\langle\nabla_y\widetilde{f}(x_t,y_t),y_t^*-y_t\right\rangle-\frac{\mu}{2}\|y_t-y_t^*\|^2\right)$$

$$\leq\sum_{t=t_0+1}^{t_1-1}\left(\frac{1-\gamma_t\mu}{2\gamma_t}\|y_t-y_t^*\|^2-\frac{1}{2\gamma_t(1+\lambda_t)}\left\|y_{t+1}-y_{t+1}^*\right\|^2\right)+\sum_{t=t_0}^{t_1-1}\frac{\gamma_t}{2}\left\|\nabla_y\widetilde{f}(x_t,y_t)\right\|^2$$

$$+\sum_{t=t_0}^{t_1-1}\frac{S_t}{2\gamma_t(1+\lambda_t)}.$$

Now we take the expectation and get

$$\mathbb{E}\left[\text{LHS}\right]\geq\mathbb{E}\left[\sum_{t=t_0}^{t_1-1}\mathbb{E}_{\xi_t^y}\left[\left(\left\langle\nabla_y\widetilde{f}(x_t,y_t),y_t^*-y_t\right\rangle-\frac{\mu}{2}\|y_t-y_t^*\|^2\right)\right]\right]$$

$$=\mathbb{E}\left[\sum_{t=t_0}^{t_1-1}\left(\langle\nabla_yf(x_t,y_t),y_t^*-y_t\rangle-\frac{\mu}{2}\|y_t-y_t^*\|^2\right)\right]$$

$$\geq\mathbb{E}\left[\sum_{t=t_0}^{t_1-1}\left(f(x_t,y_t^*)-f(x_t,y_t)\right)\right],$$

where we used strong-concavity in the last inequality.

$\square$

**Lemma C.2.** *Under the same setting as Theorem 3.2, if $v_{t+1}^y \leq C$ for $t = 0, ..., t_0 - 1$, then we have*

$$
\mathbb{E}\left[\sum_{t=0}^{t_0-1} (f(x_t, y_t^*) - f(x_t, y_t))\right]
$$

$$
\leq \mathbb{E}\left[\sum_{t=0}^{t_0-1}\left(\frac{1-\gamma_t\mu}{2\gamma_t}\|y_t - y_t^*\|^2 - \frac{1}{\gamma_t(2+\mu\gamma_t)}\left\|y_{t+1} - y_{t+1}^*\right\|^2\right)\right] + \mathbb{E}\left[\sum_{t=0}^{t_0-1}\frac{\gamma_t}{2}\left\|\nabla_y \widetilde{f}(x_t, y_t)\right\|^2\right]
$$

$$
+ \frac{\kappa^2 \left(\mu\eta^y C^\beta + 2C^{2\beta}\right)(\eta^x)^2}{2\mu(\eta^y)^2}\mathbb{E}\left[\frac{1 + \log v_{t_0}^x - \log v_0^x}{(v_0^x)^{2\alpha-1}} \cdot \mathbf{1}_{\alpha \geq 0.5} + \frac{\left(v_{t_0}^x\right)^{1-2\alpha}}{1-2\alpha} \cdot \mathbf{1}_{\alpha < 0.5}\right].
$$

*Proof.* By Young's inequality, we have

$$
\left\|y_{t+1} - y_{t+1}^*\right\|^2 \leq (1+\lambda_t)\|y_{t+1} - y_t^*\|^2 + \left(1+\frac{1}{\lambda_t}\right)\left\|y_{t+1}^* - y_t^*\right\|^2.
$$

Then letting $\lambda_t = \frac{\mu\gamma_t}{2}$ and by Lemma C.1, we have

$$
\mathbb{E}\left[\sum_{t=0}^{t_0-1}(f(x_t, y_t^*) - f(x_t, y_t))\right]
$$

$$
\leq \mathbb{E}\left[\sum_{t=0}^{t_0-1}\left(\frac{1-\gamma_t\mu}{2\gamma_t}\|y_t - y_t^*\|^2 - \frac{1}{\gamma_t(2+\mu\gamma_t)}\left\|y_{t+1} - y_{t+1}^*\right\|^2\right)\right]
$$

$$
+ \mathbb{E}\left[\sum_{t=0}^{t_0-1}\frac{\gamma_t}{2}\left\|\nabla_y \widetilde{f}(x_t, y_t)\right\|^2\right] + \mathbb{E}\left[\sum_{t=0}^{t_0-1}\frac{\left(1+\frac{2}{\mu\gamma_t}\right)}{\gamma_t(2+\mu\gamma_t)}\left\|y_{t+1}^* - y_t^*\right\|^2\right].
$$

We now remain to bound the last term:

$$
\mathbb{E}\left[\sum_{t=0}^{t_0-1}\frac{\left(1+\frac{2}{\mu\gamma_t}\right)}{\gamma_t(2+\mu\gamma_t)}\left\|y_{t+1}^* - y_t^*\right\|^2\right]
$$

$$
\leq \mathbb{E}\left[\sum_{t=0}^{t_0-1}\frac{\left(1+\frac{2}{\mu\gamma_t}\right)}{2\gamma_t}\left\|y_{t+1}^* - y_t^*\right\|^2\right]
$$

$$
= \mathbb{E}\left[\sum_{t=0}^{t_0-1}\frac{\mu\eta^y\left(v_{t+1}^y\right)^\beta + 2\left(v_{t+1}^y\right)^{2\beta}}{2\mu(\eta^y)^2}\left\|y_{t+1}^* - y_t^*\right\|^2\right]
$$

$$
\leq \frac{\mu\eta^y C^\beta + 2C^{2\beta}}{2\mu(\eta^y)^2}\mathbb{E}\left[\sum_{t=0}^{t_0-1}\left\|y_{t+1}^* - y_t^*\right\|^2\right].
$$

By Lemma B.2 we have

$$
\sum_{t=0}^{t_0-1}\left\|y_{t+1}^* - y_t^*\right\|^2 \leq \kappa^2\sum_{t=0}^{t_0-1}\|x_{t+1} - x_t\|^2
$$

$$
= \kappa^2\sum_{t=0}^{t_0-1}\eta_t^2\left\|\nabla_x \widetilde{f}(x_t, y_t)\right\|^2
$$

$$
= \kappa^2(\eta^x)^2\sum_{t=0}^{t_0-1}\frac{1}{\max\left\{v_{t+1}^x, v_{t+1}^y\right\}^{2\alpha}}\left\|\nabla_x \widetilde{f}(x_t, y_t)\right\|^2
$$

$$
\leq \kappa^2(\eta^x)^2\sum_{t=0}^{t_0-1}\frac{1}{\left(v_{t+1}^x\right)^{2\alpha}}\left\|\nabla_x \widetilde{f}(x_t, y_t)\right\|^2
$$

$$\leq \kappa^2 (\eta^x)^2 \left( \frac{v_0^x}{(v_0^x)^{2\alpha}} + \sum_{t=0}^{t_0-1} \frac{1}{(v_{t+1}^x)^{2\alpha}} \left\| \nabla_x \widetilde{f}(x_t, y_t) \right\|^2 \right)$$

$$\leq \kappa^2 (\eta^x)^2 \left( \frac{1 + \log v_{t_0}^x - \log v_0^x}{(v_0^x)^{2\alpha-1}} \cdot \mathbf{1}_{\alpha \geq 0.5} + \frac{(v_{t_0}^x)^{1-2\alpha}}{1-2\alpha} \cdot \mathbf{1}_{\alpha < 0.5} \right)$$

where we applied Lemma B.1 in the last inequality. Bringing back this result, we finish the proof.

□

**Lemma C.3.** *Under the same setting as Theorem 3.2, if $t_0$ is the first iteration such that $v_{t_0+1}^y > C$, then we have*

$$\mathbb{E} \left[ \sum_{t=t_0}^{T-1} (f(x_t, y_t^*) - f(x_t, y_t)) \right]$$

$$\leq \mathbb{E} \left[ \sum_{t=t_0}^{T-1} \left( \frac{1-\gamma_t\mu}{2\gamma_t} \|y_t - y_t^*\|^2 - \frac{1}{\gamma_t(2+\mu\gamma_t)} \left\| y_{t+1} - y_{t+1}^* \right\|^2 \right) \right] + \mathbb{E} \left[ \sum_{t=t_0}^{T-1} \frac{\gamma_t}{2} \left\| \nabla_y \widetilde{f}(x_t, y_t) \right\|^2 \right]$$

$$+ \left( \kappa^2 + \frac{\widehat{L}^2 G^2 (\eta^x)^2}{\mu \eta^y (v_0^y)^{2\alpha-\beta}} \right) \frac{(\eta^x)^2}{2(1-\alpha)\eta^y (v_0^y)^{\alpha-\beta}} \mathbb{E} \left[ (v_T^x)^{1-\alpha} \right]$$

$$+ \frac{2\kappa^2 (\eta^x)^2}{\mu (\eta^y)^2 C^{2\alpha-2\beta}} \mathbb{E} \left[ \sum_{t=t_0}^{T-1} \|\nabla_x f(x_t, y_t)\|^2 \right] + \left( \frac{1}{\mu} + \frac{\eta^y}{(v_0^y)^\beta} \right) \frac{4\kappa\eta^x G^2}{\eta^y (v_0^y)^\alpha} \mathbb{E} \left[ (v_T^y)^\beta \right].$$

*Proof.* By the Lipschitzness of $y^*(\cdot)$ as in Lemma B.2, we have

$$\left\| y_{t+1} - y_{t+1}^* \right\|^2 = \|y_{t+1} - y_t^*\|^2 + \left\| y_t^* - y_{t+1}^* \right\|^2 + 2\langle y_{t+1} - y_t^*, y_t^* - y_{t+1}^* \rangle$$

$$\leq \|y_{t+1} - y_t^*\|^2 + \kappa^2 \eta_t^2 \left\| \nabla_x \widetilde{f}(x_t, y_t) \right\|^2 + 2\langle y_{t+1} - y_t^*, y_t^* - y_{t+1}^* \rangle$$

$$\leq \|y_{t+1} - y_t^*\|^2 + \kappa^2 \eta_t^2 \left\| \nabla_x \widetilde{f}(x_t, y_t) \right\|^2 \underbrace{-2 (y_{t+1} - y_t^*)^\mathsf{T} \nabla y^*(x_t) (x_{t+1} - x_t)}_{(C)}$$

$$+ \underbrace{2 (y_{t+1} - y_t^*)^\mathsf{T} \left( y_t^* - y_{t+1}^* + \nabla y^*(x_t) (x_{t+1} - x_t) \right)}_{(D)}.$$

For Term (C), by the Cauchy-Schwarz and Lipschitzness of $y^*(\cdot)$,

$$- 2 (y_{t+1} - y_t^*)^\mathsf{T} \nabla y^*(x_t) (x_{t+1} - x_t)$$

$$= 2\eta_t (y_{t+1} - y_t^*)^\mathsf{T} \nabla y^*(x_t)\nabla_x f(x_t, y_t) + 2\eta_t (y_{t+1} - y_t^*)^\mathsf{T} \nabla y^*(x_t) \left( \nabla_x \widetilde{f}(x_t, y_t) - \nabla_x f(x_t, y_t) \right)$$

$$\leq 2\eta_t \|y_{t+1} - y_t^*\| \|\nabla y^*(x_t)\| \|\nabla_x f(x_t, y_t)\| + 2\eta_t (y_{t+1} - y_t^*)^\mathsf{T} \nabla y^*(x_t) \left( \nabla_x \widetilde{f}(x_t, y_t) - \nabla_x f(x_t, y_t) \right)$$

$$\leq 2\|y_{t+1} - y_t^*\| \kappa\eta_t \|\nabla_x f(x_t, y_t)\| + 2\eta_t (y_{t+1} - y_t^*)^\mathsf{T} \nabla y^*(x_t) \left( \nabla_x \widetilde{f}(x_t, y_t) - \nabla_x f(x_t, y_t) \right)$$

$$\leq \lambda_t \|y_{t+1} - y_t^*\|^2 + \frac{\kappa^2 \eta_t^2}{\lambda_t} \|\nabla_x f(x_t, y_t)\|^2 + 2\eta_t (y_{t+1} - y_t^*)^\mathsf{T} \nabla y^*(x_t) \left( \nabla_x \widetilde{f}(x_t, y_t) - \nabla_x f(x_t, y_t) \right),$$

where we used Young's inequality in the last step and $\lambda_t > 0$ will be determined later.

For Term (D), according to Cauchy-Schwarz and the smoothness of $y^*(\cdot)$ as shown in Lemma B.3,

$$2 (y_{t+1} - y_t^*)^\mathsf{T} \left( y_t^* - y_{t+1}^* + \nabla y^*(x_t) (x_{t+1} - x_t) \right)$$

$$\leq 2\|y_{t+1} - y_t^*\| \left\| y_t^* - y_{t+1}^* + \nabla y^*(x_t) (x_{t+1} - x_t) \right\|$$

$$\leq 2\|y_{t+1} - y_t^*\| \cdot \frac{\widehat{L}}{2} \|x_{t+1} - x_t\|^2$$

$$= \widehat{L}\eta_t^2 \|y_{t+1} - y_t^*\| \left\|\nabla_x \widetilde{f}(x_t, y_t)\right\|^2$$

$$\leq \widehat{L}\eta_t^2 \|y_{t+1} - y_t^*\| \|G \cdot \left\|\nabla_x \widetilde{f}(x_t, y_t)\right\|$$

$$\leq \frac{\tau \widehat{L} G^2 \eta_t^2}{2} \|y_{t+1} - y_t^*\|^2 + \frac{\widehat{L}\eta_t^2}{2\tau} \left\|\nabla_x \widetilde{f}(x_t, y_t)\right\|^2,$$

where in the last step we used Young's inequality and $\tau > 0$.

Therefore, in total, we have

$$\left\|y_{t+1} - y_{t+1}^*\right\|^2 \leq \left(1 + \lambda_t + \frac{\tau \widehat{L} G^2 \eta_t^2}{2}\right) \|y_{t+1} - y_t^*\|^2 + \left(\kappa^2 + \frac{\widehat{L}}{2\tau}\right) \eta_t^2 \left\|\nabla_x \widetilde{f}(x_t, y_t)\right\|^2$$

$$+ \frac{\kappa^2 \eta_t^2}{\lambda_t} \|\nabla_x f(x_t, y_t)\|^2 + 2\eta_t (y_{t+1} - y_t^*)^\mathsf{T} \nabla y^*(x_t) \left(\nabla_x \widetilde{f}(x_t, y_t) - \nabla_x f(x_t, y_t)\right).$$

Note that we can upper bound $\eta_t$ by

$$\eta_t = \frac{\eta^x}{\max\left\{v_{t+1}^x, v_{t+1}^y\right\}^\alpha} \leq \frac{\eta^x}{\left(v_{t+1}^y\right)^\alpha} \leq \frac{\eta^x}{\left(v_0^y\right)^\alpha},$$

and

$$\eta_t \leq \frac{\eta^x}{\left(v_{t+1}^y\right)^\alpha} = \frac{\eta^x}{\left(v_{t+1}^y\right)^{\alpha-\beta}\left(v_{t+1}^y\right)^\beta} \leq \frac{\eta^x}{\left(v_0^y\right)^{\alpha-\beta}\left(v_{t+1}^y\right)^\beta},$$

which, plugged into the previous result, implies

$$\left\|y_{t+1} - y_{t+1}^*\right\|^2 \leq \left(1 + \lambda_t + \frac{\tau \widehat{L} G^2 (\eta^x)^2}{2\left(v_0^y\right)^{2\alpha-\beta}\left(v_{t+1}^y\right)^\beta}\right) \|y_{t+1} - y_t^*\|^2 + \left(\kappa^2 + \frac{\widehat{L}}{2\tau}\right) \eta_t^2 \left\|\nabla_x \widetilde{f}(x_t, y_t)\right\|^2$$

$$+ \frac{\kappa^2 \eta_t^2}{\lambda_t} \|\nabla_x f(x_t, y_t)\|^2 + 2\eta_t (y_{t+1} - y_t^*)^\mathsf{T} \nabla y^*(x_t) \left(\nabla_x \widetilde{f}(x_t, y_t) - \nabla_x f(x_t, y_t)\right).$$

Now we choose $\lambda_t = \frac{\mu\eta^y}{4\left(v_{t+1}^y\right)^\beta}$ and $\tau = \frac{\mu\eta^y\left(v_0^y\right)^{2\alpha-\beta}}{2\widehat{L} G^2 (\eta^x)^2}$, and get

$$\left\|y_{t+1} - y_{t+1}^*\right\|^2$$

$$\leq \left(1 + \frac{\mu\eta^y}{2\left(v_{t+1}^y\right)^\beta}\right) \|y_{t+1} - y_t^*\|^2 + \left(\kappa^2 + \frac{\widehat{L}^2 G^2 (\eta^x)^2}{\mu\eta^y \left(v_0^y\right)^{2\alpha-\beta}}\right) \eta_t^2 \left\|\nabla_x \widetilde{f}(x_t, y_t)\right\|^2$$

$$+ \frac{4\kappa^2 \left(v_{t+1}^y\right)^\beta \eta_t^2}{\mu\eta^y} \|\nabla_x f(x_t, y_t)\|^2 + 2\eta_t (y_{t+1} - y_t^*)^\mathsf{T} \nabla y^*(x_t) \left(\nabla_x \widetilde{f}(x_t, y_t) - \nabla_x f(x_t, y_t)\right).$$

Then Lemma C.1 gives us

$$\mathbb{E}\left[\sum_{t=t_0}^{T-1} (f(x_t, y_t^*) - f(x_t, y_t))\right]$$

$$\leq \mathbb{E}\left[\sum_{t=t_0}^{T-1} \left(\frac{1 - \gamma_t \mu}{2\gamma_t} \|y_t - y_t^*\|^2 - \frac{1}{\gamma_t(2 + \mu\gamma_t)} \left\|y_{t+1} - y_{t+1}^*\right\|^2\right)\right] + \mathbb{E}\left[\sum_{t=t_0}^{T-1} \frac{\gamma_t}{2} \left\|\nabla_y \widetilde{f}(x_t, y_t)\right\|^2\right]$$

$$+ \underbrace{\mathbb{E}\left[\sum_{t=t_0}^{T-1} \frac{1}{\gamma_t(2 + \mu\gamma_t)} \left(\kappa^2 + \frac{\widehat{L}^2 G^2 (\eta^x)^2}{\mu\eta^y \left(v_0^y\right)^{2\alpha-\beta}}\right) \eta_t^2 \left\|\nabla_x \widetilde{f}(x_t, y_t)\right\|^2\right]}_{(E)}$$

$$+ \underbrace{\mathbb{E}\left[\sum_{t=t_0}^{T-1} \frac{4\kappa^2 \left(v_{t+1}^y\right)^\beta \eta_t^2}{\gamma_t(2 + \mu\gamma_t)\mu\eta^y} \|\nabla_x f(x_t, y_t)\|^2\right]}_{(F)}$$

$$+ \mathbb{E}\left[\underbrace{\sum_{t=t_0}^{T-1} \frac{2\eta_t}{\gamma_t(2+\mu\gamma_t)}\left(y_{t+1}-y_t^*\right)^{\mathsf{T}}\nabla y^*(x_t)\left(\nabla_x\widetilde{f}(x_t,y_t)-\nabla_x f(x_t,y_t)\right)}_{(G)}\right]$$

Now we proceed to bound each term.

**Term (E)**

$$\begin{aligned}
\text{Term (E)} &\leq \left(\kappa^2 + \frac{\widehat{L}^2 G^2\left(\eta^x\right)^2}{\mu\eta^y\left(v_0^y\right)^{2\alpha-\beta}}\right)\mathbb{E}\left[\sum_{t=t_0}^{T-1}\frac{\eta_t^2}{2\gamma_t}\left\|\nabla_x\widetilde{f}(x_t,y_t)\right\|^2\right] \\
&= \left(\kappa^2 + \frac{\widehat{L}^2 G^2\left(\eta^x\right)^2}{\mu\eta^y\left(v_0^y\right)^{2\alpha-\beta}}\right)\mathbb{E}\left[\sum_{t=t_0}^{T-1}\frac{\left(\eta^x\right)^2\left(v_{t+1}^y\right)^\beta}{2\eta^y\max\left\{v_{t+1}^x,v_{t+1}^y\right\}^{2\alpha}}\left\|\nabla_x\widetilde{f}(x_t,y_t)\right\|^2\right] \\
&\leq \left(\kappa^2 + \frac{\widehat{L}^2 G^2\left(\eta^x\right)^2}{\mu\eta^y\left(v_0^y\right)^{2\alpha-\beta}}\right)\mathbb{E}\left[\sum_{t=t_0}^{T-1}\frac{\left(\eta^x\right)^2\left(v_{t+1}^y\right)^\beta}{2\eta^y\left(v_{t+1}^y\right)^\beta\left(v_{t+1}^y\right)^{\alpha-\beta}\left(v_{t+1}^x\right)^\alpha}\left\|\nabla_x\widetilde{f}(x_t,y_t)\right\|^2\right] \\
&\leq \left(\kappa^2 + \frac{\widehat{L}^2 G^2\left(\eta^x\right)^2}{\mu\eta^y\left(v_0^y\right)^{2\alpha-\beta}}\right)\mathbb{E}\left[\sum_{t=t_0}^{T-1}\frac{\left(\eta^x\right)^2}{2\eta^y\left(v_0^y\right)^{\alpha-\beta}\left(v_{t+1}^x\right)^\alpha}\left\|\nabla_x\widetilde{f}(x_t,y_t)\right\|^2\right] \\
&\leq \left(\kappa^2 + \frac{\widehat{L}^2 G^2\left(\eta^x\right)^2}{\mu\eta^y\left(v_0^y\right)^{2\alpha-\beta}}\right)\mathbb{E}\left[\frac{\left(\eta^x\right)^2}{2\eta^y\left(v_0^y\right)^{\alpha-\beta}}\left(\frac{v_0^x}{\left(v_0^x\right)^\alpha}+\sum_{t=0}^{T-1}\frac{1}{\left(v_{t+1}^x\right)^\alpha}\left\|\nabla_x\widetilde{f}(x_t,y_t)\right\|^2\right)\right] \\
&\leq \left(\kappa^2 + \frac{\widehat{L}^2 G^2\left(\eta^x\right)^2}{\mu\eta^y\left(v_0^y\right)^{2\alpha-\beta}}\right)\frac{\left(\eta^x\right)^2}{2(1-\alpha)\eta^y\left(v_0^y\right)^{\alpha-\beta}}\mathbb{E}\left[\left(v_T^x\right)^{1-\alpha}\right],
\end{aligned}$$

where we used Lemma B.1 in the last step.

**Term (F)**

$$\begin{aligned}
\text{Term (F)} &\leq \mathbb{E}\left[\sum_{t=t_0}^{T-1}\frac{2\kappa^2\left(v_{t+1}^y\right)^\beta\eta_t^2}{\gamma_t\mu\eta^y}\|\nabla_x f(x_t,y_t)\|^2\right] \\
&= \frac{2\kappa^2\left(\eta^x\right)^2}{\mu\left(\eta^y\right)^2}\mathbb{E}\left[\sum_{t=t_0}^{T-1}\frac{\left(v_{t+1}^y\right)^{2\beta}}{\max\left\{v_{t+1}^x,v_{t+1}^y\right\}^{2\alpha}}\|\nabla_x f(x_t,y_t)\|^2\right] \\
&\leq \frac{2\kappa^2\left(\eta^x\right)^2}{\mu\left(\eta^y\right)^2}\mathbb{E}\left[\sum_{t=t_0}^{T-1}\frac{\left(v_{t+1}^y\right)^{2\beta}}{\left(v_{t+1}^y\right)^{2\alpha}}\|\nabla_x f(x_t,y_t)\|^2\right] \\
&\leq \frac{2\kappa^2\left(\eta^x\right)^2}{\mu\left(\eta^y\right)^2}\mathbb{E}\left[\frac{1}{\left(v_{t_0+1}^y\right)^{2\alpha-2\beta}}\sum_{t=t_0}^{T-1}\|\nabla_x f(x_t,y_t)\|^2\right] \\
&\leq \frac{2\kappa^2\left(\eta^x\right)^2}{\mu\left(\eta^y\right)^2 C^{2\alpha-2\beta}}\mathbb{E}\left[\sum_{t=t_0}^{T-1}\|\nabla_x f(x_t,y_t)\|^2\right]
\end{aligned}$$

**Term (G)** For simplicity, denote $m_t := \frac{2}{\gamma_t(2+\mu\gamma_t)}\left(y_{t+1}-y_t^*\right)^{\mathsf{T}}\nabla y^*(x_t)\left(\nabla_x\widetilde{f}(x_t,y_t)-\nabla_x f(x_t,y_t)\right)$
Since $y^*(\cdot)$ is $\kappa$-Lipschitz as in Lemma B.2, $|m_t|$ can be upper bounded as

$$\begin{aligned}
|m_t| &\leq \frac{1}{\gamma_t}\|y_{t+1}-y_t^*\|\|\nabla y^*(x_t)\|\left(\left\|\nabla_x\widetilde{f}(x_t,y_t)\right\|+\|\nabla_x f(x_t,y_t)\|\right) \\
&\leq \frac{\kappa}{\gamma_t}\|y_{t+1}-y_t^*\|\left(\left\|\nabla_x\widetilde{f}(x_t,y_t)\right\|+\|\nabla_x f(x_t,y_t)\|\right)
\end{aligned}$$

$$\leq \frac{\kappa}{\gamma_t} \left\| \mathcal{P}_{\mathcal{Y}} \left( y_t + \gamma_t \nabla_y \widetilde{f}(x_t, y_t) \right) - y_t^* \right\| \left( \left\| \nabla_x \widetilde{f}(x_t, y_t) \right\| + \| \nabla_x f(x_t, y_t) \| \right)$$

$$\leq \frac{\kappa}{\gamma_t} \left\| y_t + \gamma_t \nabla_y \widetilde{f}(x_t, y_t) - y_t^* \right\| \left( \left\| \nabla_x \widetilde{f}(x_t, y_t) \right\| + \| \nabla_x f(x_t, y_t) \| \right)$$

$$\leq \frac{\kappa}{\gamma_t} \left( \| y_t - y_t^* \| + \left\| \gamma_t \nabla_y \widetilde{f}(x_t, y_t) \right\| \right) \left( \left\| \nabla_x \widetilde{f}(x_t, y_t) \right\| + \| \nabla_x f(x_t, y_t) \| \right)$$

$$\leq \frac{\kappa}{\gamma_t} \left( \frac{1}{\mu} \| \nabla_y f(x_t, y_t) \| + \left\| \gamma_t \nabla_y \widetilde{f}(x_t, y_t) \right\| \right) \left( \left\| \nabla_x \widetilde{f}(x_t, y_t) \right\| + \| \nabla_x f(x_t, y_t) \| \right)$$

$$\leq \underbrace{\frac{2G\kappa}{\gamma_{T-1}} \left( \frac{G}{\mu} + \frac{\eta^y G}{(v_0^y)^\beta} \right)}_{M}.$$

Also note that $\gamma_t$ and $y_{t+1}$ does not depend on $\xi_t^x$, so $\mathbb{E}_{\xi_t^x}[m_t] = 0$. Next, we look at Term (G).

$$\text{Term (G)} = \mathbb{E} \left[ \sum_{t=t_0}^{T-1} \eta_t m_t \right]$$

$$= \mathbb{E} \left[ \eta_{t_0} m_{t_0} + \sum_{t=t_0+1}^{T-1} \eta_{t-1} m_t + \sum_{t=t_0+1}^{T-1} (\eta_t - \eta_{t-1}) m_t \right]$$

$$\leq \mathbb{E} \left[ \frac{\eta^x}{(v_0^y)^\alpha} M + \sum_{t=t_0+1}^{T-1} \eta_{t-1} \mathbb{E}_{\xi_t^x}[m_t] + \sum_{t=t_0+1}^{T-1} (\eta_{t-1} - \eta_t)(-m_t) \right]$$

$$\leq \mathbb{E} \left[ \frac{\eta^x}{(v_0^y)^\alpha} M + \sum_{t=t_0+1}^{T-1} (\eta_{t-1} - \eta_t) M \right]$$

$$\leq \mathbb{E} \left[ \frac{2\eta^x}{(v_0^y)^\alpha} M \right]$$

$$= \left( \frac{1}{\mu} + \frac{\eta^y}{(v_0^y)^\beta} \right) \frac{4\kappa \eta^x G^2}{\eta^y (v_0^y)^\alpha} \mathbb{E} \left[ (v_T^y)^\beta \right].$$

Summarizing all the results, we finish the proof.

$\square$

**Lemma C.4.** *Under the same setting as Theorem 3.2, we have*

$$\mathbb{E} \left[ \sum_{t=0}^{T-1} \left( \frac{1 - \gamma_t \mu}{2\gamma_t} \| y_t - y_t^* \|^2 - \frac{1}{\gamma_t(2 + \mu \gamma_t)} \| y_{t+1} - y_{t+1}^* \|^2 \right) \right]$$

$$\leq \frac{(v_0^y)^\beta G^2}{2\mu^2 \eta^y} + \frac{(2\beta G)^{\frac{1}{1-\beta}+2} G^2}{4\mu^{\frac{1}{1-\beta}+3} (\eta^y)^{\frac{1}{1-\beta}+2} (v_0^y)^{2-2\beta}}.$$

*Proof.*

$$\mathbb{E} \left[ \sum_{t=0}^{T-1} \left( \frac{1 - \gamma_t \mu}{2\gamma_t} \| y_t - y_t^* \|^2 - \frac{1}{\gamma_t(2 + \mu \gamma_t)} \left\| y_{t+1} - y_{t+1}^* \right\|^2 \right) \right]$$

$$\leq \left( \frac{(v_0^y)^\beta}{2\eta^y} - \frac{\mu}{2} \right) \| y_0 - y_0^* \|^2 + \frac{1}{2\eta^y} \sum_{t=1}^{T-1} \left( (v_{t+1}^y)^\beta - \frac{\mu \eta^y}{2} - (v_t^y)^\beta - \frac{\mu^2 (\eta^y)^2}{4 (v_t^y)^\beta + 2\mu \eta^y} \right) \| y_t - y_t^* \|^2$$

$$\leq \frac{(v_0^y)^\beta G^2}{2\mu^2 \eta^y} + \frac{1}{2\eta^y} \underbrace{\sum_{t=1}^{T-1} \left( (v_{t+1}^y)^\beta - \frac{\mu \eta^y}{2} - (v_t^y)^\beta \right) \| y_t - y_t^* \|^2}_{(H)}.$$

For Term (H), we will bound it using the same strategy as in (Yang et al., 2022a). The general idea is to show that $\left(v_{t+1}^y\right)^\beta - \frac{\mu\eta^y}{2} - (v_t^y)^\beta$ is positive for only a constant number of times. If the term is positive at iteration $t$, then we have

$$
\begin{aligned}
0 &< \left(v_{t+1}^y\right)^\beta - (v_t^y)^\beta - \frac{\mu\eta^y}{2} \\
&= \left(v_t^y + \left\|\nabla_y\widetilde{f}(x_t, y_t)\right\|^2\right)^\beta - (v_t^y)^\beta - \frac{\mu\eta^y}{2} \\
&= (v_t^y)^\beta \left(1 + \frac{\left\|\nabla_y\widetilde{f}(x_t, y_t)\right\|^2}{v_t^y}\right)^\beta - (v_t^y)^\beta - \frac{\mu\eta^y}{2} \\
&\leq (v_t^y)^\beta \left(1 + \frac{\beta\left\|\nabla_y\widetilde{f}(x_t, y_t)\right\|^2}{v_t^y}\right) - (v_t^y)^\beta - \frac{\mu\eta^y}{2} \\
&= \frac{\beta\left\|\nabla_y\widetilde{f}(x_t, y_t)\right\|^2}{(v_t^y)^{1-\beta}} - \frac{\mu\eta^y}{2},
\end{aligned}
\tag{11}
$$

where in the last inequality we used Bernoulli's inequality. By rearranging the terms, we have the two following conditions

$$
\begin{cases}
\left\|\nabla_y\widetilde{f}(x_t, y_t)\right\|^2 > \frac{\mu\eta^y}{2\beta}(v_t^y)^{1-\beta} \geq \frac{\mu\eta^y}{2\beta}(v_0^y)^{1-\beta} \\
(v_t^y)^{1-\beta} < \frac{2\beta}{\mu\eta^y}\left\|\nabla_y\widetilde{f}(x_t, y_t)\right\|^2 \leq \frac{2\beta G}{\mu\eta^y},
\end{cases}
$$

This indicates that at each time the term is positive, the gradient norm must be large enough and the accumulated gradient norm, i.e., $v_{t+1}^y$, must be small enough. Therefore, we can have at most

$$
\frac{\left(\frac{2\beta G}{\mu\eta^y}\right)^{\frac{1}{1-\beta}}}{\frac{\mu\eta^y}{2\beta}(v_0^y)^{1-\beta}}
$$

constant number of iterations when the term is positive. When the term is positive, it is also upper bounded by using the result from Equation (11):

$$
\begin{aligned}
\left(\left(v_{t+1}^y\right)^\beta - \frac{\mu\eta^y}{2} - (v_t^y)^\beta\right)\|y_t - y_t^*\|^2 &\leq \frac{\beta\left\|\nabla_y\widetilde{f}(x_t, y_t)\right\|^2}{(v_t^y)^{1-\beta}}\|y_t - y_t^*\|^2 \\
&\leq \frac{\beta G^2}{(v_0^y)^{1-\beta}}\|y_t - y_t^*\|^2 \\
&\leq \frac{\beta G^2}{\mu^2(v_0^y)^{1-\beta}}\|\nabla_y f(x_t, y_t)\|^2 \\
&\leq \frac{\beta G^4}{\mu^2(v_0^y)^{1-\beta}}
\end{aligned}
$$

which is a constant. In total, Term (H) is bounded by

$$
\frac{(2\beta G)^{\frac{1}{1-\beta}+2}G^2}{2\mu^{\frac{1}{1-\beta}+3}(\eta^y)^{\frac{1}{1-\beta}+1}(v_0^y)^{2-2\beta}}.
$$

Bringing it back, we get the desired result. $\qquad\square$

**Lemma C.5.** *Under the same setting as Theorem 3.2, for any constant $C$, we have*

$$\mathbb{E}\left[\sum_{t=0}^{T-1}(f(x_t, y_t^*) - f(x_t, y_t))\right]$$

$$\leq \frac{2\kappa^2 (\eta^x)^2}{\mu (\eta^y)^2 C^{2\alpha-2\beta}}\mathbb{E}\left[\sum_{t=0}^{T-1}\|\nabla_x f(x_t, y_t)\|^2\right] + \frac{\eta^y}{2(1-\beta)}\mathbb{E}\left[(v_T^y)^{1-\beta}\right]$$

$$+ \left(\frac{1}{\mu} + \frac{\eta^y}{(v_0^y)^\beta}\right)\frac{4\kappa\eta^x G^2}{\eta^y (v_0^y)^\alpha}\mathbb{E}\left[(v_T^y)^\beta\right]$$

$$+ \frac{\kappa^2 \left(\mu\eta^y C^\beta + 2C^{2\beta}\right)(\eta^x)^2}{2\mu (\eta^y)^2}\mathbb{E}\left[\frac{1 + \log v_T^x - \log v_0^x}{(v_0^x)^{2\alpha-1}}\cdot \mathbf{1}_{\alpha\geq 0.5} + \frac{(v_T^x)^{1-2\alpha}}{1-2\alpha}\cdot \mathbf{1}_{\alpha<0.5}\right]$$

$$+ \left(\kappa^2 + \frac{\widehat{L}^2 G^2 (\eta^x)^2}{\mu\eta^y (v_0^y)^{2\alpha-\beta}}\right)\frac{(\eta^x)^2}{2(1-\alpha)\eta^y (v_0^y)^{\alpha-\beta}}\mathbb{E}\left[(v_T^x)^{1-\alpha}\right]$$

$$+ \frac{(v_0^y)^\beta G^2}{2\mu^2\eta^y} + \frac{(2\beta G)^{\frac{1}{1-\beta}+2} G^2}{4\mu^{\frac{1}{1-\beta}+3} (\eta^y)^{\frac{1}{1-\beta}+2} (v_0^y)^{2-2\beta}}.$$

*Proof.* By Lemma C.2 and Lemma C.3, we have for any constant $C$,

$$\mathbb{E}\left[\sum_{t=0}^{T-1}(f(x_t, y_t^*) - f(x_t, y_t))\right]$$

$$\leq \mathbb{E}\left[\sum_{t=0}^{T-1}\left(\frac{1-\gamma_t\mu}{2\gamma_t}\|y_t - y_t^*\|^2 - \frac{1}{\gamma_t(2+\mu\gamma_t)}\|y_{t+1} - y_{t+1}^*\|^2\right)\right]$$

$$+ \mathbb{E}\left[\sum_{t=0}^{T-1}\frac{\gamma_t}{2}\left\|\nabla_y\widetilde{f}(x_t, y_t)\right\|^2\right] + \frac{2\kappa^2 (\eta^x)^2}{\mu (\eta^y)^2 C^{2\alpha-2\beta}}\mathbb{E}\left[\sum_{t=0}^{T-1}\|\nabla_x f(x_t, y_t)\|^2\right]$$

$$+ \frac{\kappa^2 \left(\mu\eta^y C^\beta + 2C^{2\beta}\right)(\eta^x)^2}{2\mu (\eta^y)^2}\mathbb{E}\left[\frac{1 + \log v_T^x - \log v_0^x}{(v_0^x)^{2\alpha-1}}\cdot \mathbf{1}_{\alpha\geq 0.5} + \frac{(v_T^x)^{1-2\alpha}}{1-2\alpha}\cdot \mathbf{1}_{\alpha<0.5}\right]$$

$$+ \left(\kappa^2 + \frac{\widehat{L}^2 G^2 (\eta^x)^2}{\mu\eta^y (v_0^y)^{2\alpha-\beta}}\right)\frac{(\eta^x)^2}{2(1-\alpha)\eta^y (v_0^y)^{\alpha-\beta}}\mathbb{E}\left[(v_T^x)^{1-\alpha}\right]$$

$$+ \left(\frac{1}{\mu} + \frac{\eta^y}{(v_0^y)^\beta}\right)\frac{4\kappa\eta^x G^2}{\eta^y (v_0^y)^\alpha}\mathbb{E}\left[(v_T^y)^\beta\right].$$

The first term can be bounded by Lemma C.4. For the second term, we have

$$\mathbb{E}\left[\sum_{t=0}^{T-1}\frac{\gamma_t}{2}\left\|\nabla_y\widetilde{f}(x_t, y_t)\right\|^2\right] = \mathbb{E}\left[\sum_{t=0}^{T-1}\frac{\eta^y}{2 (v_{t+1}^y)^\beta}\left\|\nabla_y\widetilde{f}(x_t, y_t)\right\|^2\right]$$

$$\leq \frac{\eta^y}{2}\mathbb{E}\left[\frac{v_0^y}{(v_0^y)^\beta} + \sum_{t=0}^{T-1}\frac{1}{(v_{t+1}^y)^\beta}\left\|\nabla_y\widetilde{f}(x_t, y_t)\right\|^2\right]$$

$$\leq \frac{\eta^y}{2(1-\beta)}\mathbb{E}\left[(v_T^y)^{1-\beta}\right],$$

where the last inequality follows from Lemma B.1. Then the proof is completed. $\square$

## C.3 PROOF OF THEOREM 3.2

We present a formal version of Theorem 3.2.

**Theorem C.2** (stochastic setting). *Under Assumptions 3.1 to 3.6, Algorithm 1 with stochastic gradient oracles satisfies that for any $0 < \beta < \alpha < 1$, after $T$ iterations,*

$$
\mathbb{E}\left[\frac{1}{T}\sum_{t=0}^{T-1}\|\nabla_x f(x_t, y_t)\|^2\right]
$$

$$
\leq \frac{4\Delta\Phi G^{2\alpha}}{\eta^x T^{1-\alpha}} + \left(\frac{4l\kappa\eta^x}{1-\alpha} + \left(\kappa^2 + \frac{\widehat{L}^2 G^2 (\eta^x)^2}{\mu\eta^y (v_0^y)^{2\alpha-\beta}}\right)\frac{2l\kappa(\eta^x)^2}{(1-\alpha)\eta^y (v_0^y)^{\alpha-\beta}}\right)\frac{G^{2(1-\alpha)}}{T^\alpha}
$$

$$
+ \frac{2l\kappa\eta^y G^{2(1-\beta)}}{(1-\beta)T^\beta} + \left(\frac{1}{\mu} + \frac{\eta^y}{(v_0^y)^\beta}\right)\frac{16l\kappa^2\eta^x G^{2(1+\beta)}}{\eta^y (v_0^y)^\alpha T^{1-\beta}}
$$

$$
+ \frac{2\kappa^4\left(\mu\eta^y C^\beta + 2C^{2\beta}\right)(\eta^x)^2}{(\eta^y)^2}\left(\frac{1 + \log(G^2 T) - \log v_0^x}{(v_0^x)^{2\alpha-1}T}\cdot\mathbf{1}_{\alpha\geq 0.5} + \frac{G^{2(1-2\alpha)}}{(1-2\alpha)T^{2\alpha}}\cdot\mathbf{1}_{\alpha<0.5}\right)
$$

$$
+ \frac{2\kappa^2 (v_0^y)^\beta G^2}{\mu\eta^y T} + \frac{l\kappa(2\beta G)^{\frac{1}{1-\beta}+2} G^2}{\mu^{\frac{1}{1-\beta}+3}(\eta^y)^{\frac{1}{1-\beta}+2}(v_0^y)^{2-2\beta}T},
$$

*and*

$$
\mathbb{E}\left[\frac{1}{T}\sum_{t=0}^{T-1}\|\nabla_y f(x_t, y_t)\|^2\right]
$$

$$
\leq \frac{4\kappa^3 (\eta^x)^2}{(\eta^y)^2 C^{2\alpha-2\beta}}\mathbb{E}\left[\frac{1}{T}\sum_{t=0}^{T-1}\|\nabla_x f(x_t, y_t)\|^2\right] + \frac{l\eta^y G^{2-2\beta}}{(1-\beta)T^\beta} + \left(\frac{1}{\mu} + \frac{\eta^y}{(v_0^y)^\beta}\right)\frac{8l\kappa\eta^x G^{2+2\beta}}{\eta^y (v_0^y)^\alpha T^{1-\beta}}
$$

$$
+ \frac{\kappa^3\left(\mu\eta^y C^\beta + 2C^{2\beta}\right)(\eta^x)^2}{(\eta^y)^2}\left(\frac{1 + \log T G^2 - \log v_0^x}{(v_0^x)^{2\alpha-1}T}\cdot\mathbf{1}_{\alpha\geq 0.5} + \frac{G^{2-4\alpha}}{(1-2\alpha)T^{2\alpha}}\cdot\mathbf{1}_{\alpha<0.5}\right)
$$

$$
+ \left(\kappa^2 + \frac{\widehat{L}^2 G^2 (\eta^x)^2}{\mu\eta^y (v_0^y)^{2\alpha-\beta}}\right)\frac{l(\eta^x)^2 G^{2-2\alpha}}{(1-\alpha)\eta^y (v_0^y)^{\alpha-\beta}T^\alpha} + \frac{\kappa(v_0^y)^\beta G^2}{\mu\eta^y T} + \frac{2l(2\beta G)^{\frac{1}{1-\beta}+2} G^2}{4\mu^{\frac{1}{1-\beta}+3}(\eta^y)^{\frac{1}{1-\beta}+2}(v_0^y)^{2-2\beta}T}.
$$

*Proof.* By smoothness of the primal function, we have

$$
\Phi(x_{t+1}) - \Phi(x_t) \leq -\eta_t\left\langle\nabla\Phi(x_t), \nabla_x\widetilde{f}(x_t, y_t)\right\rangle + l\kappa\eta_t^2\left\|\nabla_x\widetilde{f}(x_t, y_t)\right\|^2.
$$

By multiplying $\frac{1}{\eta_t}$ on both sides and taking the expectation w.r.t. the noise of current iteration, we have

$$
\mathbb{E}\left[\frac{\Phi(x_{t+1}) - \Phi(x_t)}{\eta_t}\right]
$$

$$
\leq -\langle\nabla\Phi(x_t), \nabla_x f(x_t, y_t)\rangle + l\kappa\mathbb{E}\left[\eta_t\left\|\nabla_x\widetilde{f}(x_t, y_t)\right\|^2\right]
$$

$$
= -\|\nabla_x f(x_t, y_t)\|^2 + \langle\nabla_x f(x_t, y_t) - \nabla\Phi(x_t), \nabla_x f(x_t, y_t)\rangle + l\kappa\mathbb{E}\left[\eta_t\left\|\nabla_x\widetilde{f}(x_t, y_t)\right\|^2\right]
$$

$$
\leq -\|\nabla_x f(x_t, y_t)\|^2 + \frac{1}{2}\|\nabla_x f(x_t, y_t) - \nabla\Phi(x_t)\|^2 + \frac{1}{2}\|\nabla_x f(x_t, y_t)\|^2 + l\kappa\mathbb{E}\left[\eta_t\left\|\nabla_x\widetilde{f}(x_t, y_t)\right\|^2\right]
$$

$$
= -\frac{1}{2}\|\nabla_x f(x_t, y_t)\|^2 + \frac{1}{2}\|\nabla_x f(x_t, y_t) - \nabla\Phi(x_t)\|^2 + l\kappa\mathbb{E}\left[\eta_t\left\|\nabla_x\widetilde{f}(x_t, y_t)\right\|^2\right]
$$

Summing over $t = 0$ to $T - 1$, rearranging and taking total expectation, we get

$$
\mathbb{E}\left[\sum_{t=0}^{T-1}\|\nabla_x f(x_t, y_t)\|^2\right]
$$

$$
\leq \underbrace{2\mathbb{E}\left[\sum_{t=0}^{T-1}\frac{\Phi(x_t)-\Phi(x_{t+1})}{\eta_t}\right]}_{\text{(I)}} + \underbrace{2l\kappa\mathbb{E}\left[\sum_{t=0}^{T-1}\eta_t\left\|\nabla_x\widetilde{f}(x_t,y_t)\right\|^2\right]}_{\text{(J)}} + \underbrace{\mathbb{E}\left[\sum_{t=0}^{T-1}\|\nabla_x f(x_t,y_t)-\nabla\Phi(x_t)\|^2\right]}_{\text{(K)}}.
$$

$$(12)$$

**Term (I)**

$$
2\mathbb{E}\left[\sum_{t=0}^{T-1}\frac{\Phi(x_t)-\Phi(x_{t+1})}{\eta_t}\right] \leq 2\mathbb{E}\left[\frac{\Phi(x_0)}{\eta_0}-\frac{\Phi(x_T)}{\eta_{T-1}}+\sum_{t=1}^{T-1}\Phi(x_t)\left(\frac{1}{\eta_t}-\frac{1}{\eta_{t-1}}\right)\right]
$$

$$
\leq 2\mathbb{E}\left[\frac{\Phi_{\max}}{\eta_0}-\frac{\Phi^*}{\eta_{T-1}}+\sum_{t=1}^{T-1}\Phi_{\max}\left(\frac{1}{\eta_t}-\frac{1}{\eta_{t-1}}\right)\right]
$$

$$
= 2\mathbb{E}\left[\frac{\Delta\Phi}{\eta_{T-1}}\right] = 2\mathbb{E}\left[\frac{\Delta\Phi}{\eta^x}\max\{v_T^x,v_T^y\}^\alpha\right].
$$

**Term (J)**

$$
2l\kappa\sum_{t=0}^{T-1}\mathbb{E}\left[\eta_t\left\|\nabla_x\widetilde{f}(x_t,y_t)\right\|^2\right] = 2l\kappa\mathbb{E}\left[\sum_{t=0}^{T-1}\frac{\eta^x}{\max\left\{v_{t+1}^x,v_{t+1}^y\right\}^\alpha}\left\|\nabla_x\widetilde{f}(x_t,y_t)\right\|^2\right]
$$

$$
\leq 2l\kappa\eta^x\mathbb{E}\left[\sum_{t=0}^{T-1}\frac{1}{\left(v_{t+1}^x\right)^\alpha}\left\|\nabla_x\widetilde{f}(x_t,y_t)\right\|^2\right]
$$

$$
\leq 2l\kappa\eta^x\mathbb{E}\left[\left(\frac{v_0^x}{(v_0^x)^\alpha}+\sum_{t=0}^{T-1}\frac{1}{\left(v_{t+1}^x\right)^\alpha}\left\|\nabla_x\widetilde{f}(x_t,y_t)\right\|^2\right)\right]
$$

$$
\leq \frac{2l\kappa\eta^x}{1-\alpha}\mathbb{E}\left[(v_T^x)^{1-\alpha}\right].
$$

**Term (K)**    According to the smoothness of $f(x_t,\cdot)$, we have

$$
\mathbb{E}\left[\sum_{t=0}^{T-1}\|\nabla_x f(x_t,y_t)-\nabla\Phi(x_t)\|^2\right] \leq l^2\mathbb{E}\left[\sum_{t=0}^{T-1}\|y_t-y_t^*\|^2\right] \leq 2l\kappa\mathbb{E}\left[\sum_{t=0}^{T-1}\left(f(x_t,y_t^*)-f(x_t,y_t)\right)\right],
$$

where the last inequality follows the strong-concavity of $y$. Now we let

$$
C = \left(\frac{8l\kappa^3\left(\eta^x\right)^2}{\mu\left(\eta^y\right)^2}\right)^{\frac{1}{2\alpha-2\beta}},
$$

and apply Lemma C.5, in total, we have

$$
\mathbb{E}\left[\sum_{t=0}^{T-1}\|\nabla_x f(x_t,y_t)\|^2\right]
$$

$$
\leq \frac{1}{2}\mathbb{E}\left[\sum_{t=0}^{T-1}\|\nabla_x f(x_t,y_t)\|^2\right] + 2\mathbb{E}\left[\frac{\Delta\Phi}{\eta^x}\max\{v_T^x,v_T^y\}^\alpha\right] + \frac{2l\kappa\eta^x}{1-\alpha}\mathbb{E}\left[(v_T^x)^{1-\alpha}\right]
$$

$$
+ \frac{l\kappa\eta^y}{1-\beta}\mathbb{E}\left[(v_T^y)^{1-\beta}\right] + \left(\frac{1}{\mu}+\frac{\eta^y}{(v_0^y)^\beta}\right)\frac{8l\kappa^2\eta^x G^2}{\eta^y\left(v_0^y\right)^\alpha}\mathbb{E}\left[(v_T^y)^\beta\right]
$$

$$
+ \frac{\kappa^4\left(\mu\eta^y C^\beta+2C^{2\beta}\right)\left(\eta^x\right)^2}{\left(\eta^y\right)^2}\mathbb{E}\left[\frac{1+\log v_T^x-\log v_0^x}{\left(v_0^x\right)^{2\alpha-1}}\cdot\mathbf{1}_{\alpha\geq0.5}+\frac{\left(v_T^x\right)^{1-2\alpha}}{1-2\alpha}\cdot\mathbf{1}_{\alpha<0.5}\right]
$$

$$
+ \left(\kappa^2+\frac{\widehat{L}^2 G^2\left(\eta^x\right)^2}{\mu\eta^y\left(v_0^y\right)^{2\alpha-\beta}}\right)\frac{l\kappa\left(\eta^x\right)^2}{(1-\alpha)\eta^y\left(v_0^y\right)^{\alpha-\beta}}\mathbb{E}\left[(v_T^x)^{1-\alpha}\right] + \frac{\kappa^2\left(v_0^y\right)^\beta G^2}{\mu\eta^y}
$$

$$+ \frac{l\kappa \left(2\beta G\right)^{\frac{1}{1-\beta}+2} G^2}{2\mu^{\frac{1}{1-\beta}+3} \left(\eta^y\right)^{\frac{1}{1-\beta}+2} \left(v_0^y\right)^{2-2\beta}}.$$

It remains to bound $(v_T^z)^m$ for $z \in \{x, y\}$ and $m \geq 0$:

$$(v_T^z)^m \leq \left(TG^2\right)^m.$$

Bringing it back, we conclude our proof.

$\square$

### C.4 TiAda without Accessing Opponent's Gradients

The effective stepsize of $x$ requires the knowledge of gradients of $y$, i.e., $v_{t+1}^y$. At the end of Section 3, we discussed the situation when such information is not available. Now we formally introduce the algorithm and present the convergence result.

---

**Algorithm 2** TiAda without MAX

1: **Input:** $(x_0, y_0)$, $v_0^x > 0$, $v_0^y > 0$, $\eta^x > 0$, $\eta^y > 0$, $\alpha > 0$, $\beta > 0$ and $\alpha > \beta$.
2: **for** $t = 0, 1, 2, ...$ **do**
3:    sample i.i.d. $\xi_t^x$ and $\xi_t^y$, and let $g_t^x = \nabla_x F(x_t, y_t; \xi_t^x)$ and $g_t^y = \nabla_y F(x_t, y_t; \xi_t^y)$
4:    $v_{t+1}^x = v_t^x + \|g_t^x\|^2$ and $v_{t+1}^y = v_t^y + \|g_t^y\|^2$
5:    $x_{t+1} = x_t - \frac{\eta^x}{(v_{t+1}^x)^\alpha} g_t^x$ and $y_{t+1} = \mathcal{P}_{\mathcal{Y}} \left( y_t + \frac{\eta^y}{(v_{t+1}^y)^\beta} g_t^y \right)$
6: **end for**

---

**Theorem C.3** (stochastic). *Under Assumptions 3.1, 3.2, 3.4 and 3.5, Algorithm 2 with stochastic gradient oracles satisfies that for any $0 < \beta < \alpha < 1$, after $T$ iterations,*

$$\mathbb{E}\left[\frac{1}{T}\sum_{t=0}^{T-1}\|\nabla_x f(x_t, y_t)\|^2\right]$$

$$\leq \frac{2\Delta\Phi G^{2\alpha}}{\eta^x T^{1-\alpha}} + \frac{2l\kappa\eta^x G^{2-2\alpha}}{(1-\alpha)T^\alpha} + \left(\frac{(v_0^y)^\beta G^2}{2\mu^2\eta^y} + \frac{(2\beta G)^{\frac{1}{1-\beta}+2} G^2}{4\mu^{\frac{1}{1-\beta}+3}(\eta^y)^{\frac{1}{1-\beta}+2}(v_0^y)^{2-2\beta}}\right)\frac{1}{T} + \frac{\eta^y G^{2-2\beta}}{2(1-\beta)T^\beta}$$

$$+ \left(\frac{(\eta^x)^2 \kappa^2}{2(v_0^y)^\beta \eta^y} + \frac{(\eta^x)^2 \kappa^2}{\mu(\eta^y)^2}\right)\left(\frac{\left(1+\log G^2 T - \log v_0^x\right)G^{4\beta}}{(v_0^x)^{2\alpha-1}T^{1-2\beta}}\cdot \mathbf{1}_{\alpha\geq 0.5} + \frac{G^{2-4\alpha+4\beta}}{(1-2\alpha)T^{2\alpha-2\beta}}\cdot \mathbf{1}_{\alpha<0.5}\right),$$

*and*

$$\mathbb{E}\left[\frac{1}{T}\sum_{t=0}^{T-1}\|\nabla_y f(x_t, y_t)\|^2\right]$$

$$\leq \left(\frac{\kappa(v_0^y)^\beta G^2}{\mu\eta^y} + \frac{2l(2\beta G)^{\frac{1}{1-\beta}+2} G^2}{4\mu^{\frac{1}{1-\beta}+3}(\eta^y)^{\frac{1}{1-\beta}+2}(v_0^y)^{2-2\beta}}\right)\frac{1}{T} + \frac{l\eta^y G^{2-2\beta}}{(1-\beta)T^\beta}$$

$$+ \left(\frac{l(\eta^x)^2 \kappa^2}{(v_0^y)^\beta \eta^y} + \frac{2(\eta^x)^2 \kappa^3}{(\eta^y)^2}\right)\left(\frac{\left(1+\log G^2 T - \log v_0^x\right)G^{4\beta}}{(v_0^x)^{2\alpha-1}T^{1-2\beta}}\cdot \mathbf{1}_{\alpha\geq 0.5} + \frac{G^{2-4\alpha+4\beta}}{(1-2\alpha)T^{2\alpha-2\beta}}\cdot \mathbf{1}_{\alpha<0.5}\right).$$

*Remark* C.1. The best rate achievable is $\widetilde{\mathcal{O}}\left(\epsilon^{-6}\right)$ by choosing $\alpha = 1/2$ and $\beta = 1/3$.

*Proof.* Lemmas C.1 and C.4 can be directly used here because they do not have or expand the effective stepsize of $x$, i.e., $\eta_t$. This is also the case for the beginning part of Appendix C.3, the proof of Theorem 3.2, up to Equation (12). However, we need to bound Terms (I), (J) and (K) in Equation (12) differently. According to our assumption on bounded stochastic gradients, we know that $v_T^x$ and $v_T^y$ are both upper bounded by $TG^2$, which we will use throughout the proof.

**Term (I)**

$$2\mathbb{E}\left[\sum_{t=0}^{T-1}\frac{\Phi(x_t)-\Phi(x_{t+1})}{\eta_t}\right] \le 2\mathbb{E}\left[\frac{\Phi(x_0)}{\eta_0}-\frac{\Phi(x_T)}{\eta_{T-1}}+\sum_{t=1}^{T-1}\Phi(x_t)\left(\frac{1}{\eta_t}-\frac{1}{\eta_{t-1}}\right)\right]$$

$$\le 2\mathbb{E}\left[\frac{\Phi_{\max}}{\eta_0}-\frac{\Phi^*}{\eta_{T-1}}+\sum_{t=1}^{T-1}\Phi_{\max}\left(\frac{1}{\eta_t}-\frac{1}{\eta_{t-1}}\right)\right]$$

$$=2\mathbb{E}\left[\frac{\Delta\Phi}{\eta_{T-1}}\right]=2\mathbb{E}\left[\frac{\Delta\Phi}{\eta^x}\left(v_T^x\right)^\alpha\right]\le\frac{2\Delta\Phi G^{2\alpha}T^\alpha}{\eta^x}.$$

**Term (J)**

$$2l\kappa\sum_{t=0}^{T-1}\mathbb{E}\left[\eta_t\left\|\nabla_x\widetilde{f}(x_t,y_t)\right\|^2\right]=2l\kappa\eta^x\mathbb{E}\left[\sum_{t=0}^{T-1}\frac{1}{\left(v_{t+1}^x\right)^\alpha}\left\|\nabla_x\widetilde{f}(x_t,y_t)\right\|^2\right]$$

$$\le 2l\kappa\eta^x\mathbb{E}\left[\left(\frac{v_0^x}{(v_0^x)^\alpha}+\sum_{t=0}^{T-1}\frac{1}{\left(v_{t+1}^x\right)^\alpha}\left\|\nabla_x\widetilde{f}(x_t,y_t)\right\|^2\right)\right]$$

$$\le\frac{2l\kappa\eta^x}{1-\alpha}\mathbb{E}\left[(v_T^x)^{1-\alpha}\right]\le\frac{2l\kappa\eta^x G^{2-2\alpha}T^{1-\alpha}}{1-\alpha}.$$

**Term (K)** According to the smoothness and strong-concavity of $f(x_t,\cdot)$, we have

$$\mathbb{E}\left[\sum_{t=0}^{T-1}\|\nabla_x f(x_t,y_t)-\nabla\Phi(x_t)\|^2\right]\le l^2\mathbb{E}\left[\sum_{t=0}^{T-1}\|y_t-y_t^*\|^2\right]\le 2l\kappa\mathbb{E}\left[\sum_{t=0}^{T-1}\left(f(x_t,y_t^*)-f(x_t,y_t)\right)\right].$$

To bound the RHS, we use Young's inequality and have

$$\left\|y_{t+1}-y_{t+1}^*\right\|^2\le(1+\lambda_t)\|y_{t+1}-y_t^*\|^2+\left(1+\frac{1}{\lambda_t}\right)\left\|y_{t+1}^*-y_t^*\right\|^2.$$

Then applying Lemma C.1 with $\lambda_t=\frac{\mu\gamma_t}{2}$ gives us

$$\mathbb{E}\left[\sum_{t=0}^{T-1}\left(f(x_t,y_t^*)-f(x_t,y_t)\right)\right]$$

$$\le\mathbb{E}\left[\sum_{t=0}^{T-1}\left(\frac{1-\gamma_t\mu}{2\gamma_t}\|y_t-y_t^*\|^2-\frac{1}{\gamma_t(2+\mu\gamma_t)}\left\|y_{t+1}-y_{t+1}^*\right\|^2\right)\right]$$

$$+\underbrace{\mathbb{E}\left[\sum_{t=0}^{T-1}\frac{\gamma_t}{2}\left\|\nabla_y\widetilde{f}(x_t,y_t)\right\|^2\right]}_{(L)}+\underbrace{\mathbb{E}\left[\sum_{t=0}^{T-1}\frac{\left(1+\frac{2}{\mu\gamma_t}\right)}{\gamma_t(2+\mu\gamma_t)}\left\|y_{t+1}^*-y_t^*\right\|^2\right]}_{(M)},$$

where the first term is $\mathcal{O}(1)$ according to Lemma C.4. The other two terms can be bounded as follow.

Term (L)

$$\le\mathbb{E}\left[\frac{\eta^y}{2}\left(\frac{v_0^y}{(v_0^y)^\beta}+\sum_{t=0}^{T-1}\frac{1}{\left(v_{t+1}^y\right)^\beta}\left\|\nabla_y\widetilde{f}(x_t,y_t)\right\|^2\right)\right]\le\mathbb{E}\left[\frac{\eta^y}{2(1-\beta)}\left(v_T^y\right)^{1-\beta}\right]\le\frac{\eta^y G^{2-2\beta}T^{1-\beta}}{2(1-\beta)}.$$

Term (M)

$$=\mathbb{E}\left[\sum_{t=0}^{T-1}\left(\frac{1}{\left(v_{t+1}^y\right)^\beta}+\frac{2}{\mu\eta^y}\right)\frac{\left(v_{t+1}^y\right)^{2\beta}}{2\eta^y(1+\lambda_t)}\left\|y_{t+1}^*-y_t^*\right\|^2\right]$$

$$\leq \left( \frac{1}{2\left(v_0^y\right)^\beta \eta^y} + \frac{1}{\mu(\eta^y)^2} \right) \mathbb{E}\left[ \sum_{t=0}^{T-1} \left(v_{t+1}^y\right)^{2\beta} \left\| y_{t+1}^* - y_t^* \right\|^2 \right]$$

$$\leq \left( \frac{1}{2\left(v_0^y\right)^\beta \eta^y} + \frac{1}{\mu(\eta^y)^2} \right) \mathbb{E}\left[ \left(v_T^y\right)^{2\beta} \sum_{t=0}^{T-1} \left\| y_{t+1}^* - y_t^* \right\|^2 \right]$$

$$\leq \left( \frac{\kappa^2}{2\left(v_0^y\right)^\beta \eta^y} + \frac{\kappa^2}{\mu(\eta^y)^2} \right) \mathbb{E}\left[ \left(v_T^y\right)^{2\beta} \sum_{t=0}^{T-1} \left\| x_{t+1} - x_t \right\|^2 \right]$$

$$= \left( \frac{(\eta^x)^2 \kappa^2}{2\left(v_0^y\right)^\beta \eta^y} + \frac{(\eta^x)^2 \kappa^2}{\mu(\eta^y)^2} \right) \mathbb{E}\left[ \left(v_T^y\right)^{2\beta} \sum_{t=0}^{T-1} \frac{1}{\left(v_{t+1}^x\right)^{2\alpha}} \left\| \nabla_x \widetilde{f}(x_t, y_t) \right\|^2 \right]$$

$$\leq \left( \frac{(\eta^x)^2 \kappa^2}{2\left(v_0^y\right)^\beta \eta^y} + \frac{(\eta^x)^2 \kappa^2}{\mu(\eta^y)^2} \right) \mathbb{E}\left[ \left(v_T^y\right)^{2\beta} \left( \frac{1 + \log v_T^x - \log v_0^x}{\left(v_0^x\right)^{2\alpha-1}} \cdot \mathbf{1}_{\alpha \geq 0.5} + \frac{\left(v_T^x\right)^{1-2\alpha}}{1-2\alpha} \cdot \mathbf{1}_{\alpha < 0.5} \right) \right]$$

$$\leq \left( \frac{(\eta^x)^2 \kappa^2}{2\left(v_0^y\right)^\beta \eta^y} + \frac{(\eta^x)^2 \kappa^2}{\mu(\eta^y)^2} \right) \left( \frac{\left(1 + \log G^2 T - \log v_0^x\right) G^{4\beta} T^{2\beta}}{\left(v_0^x\right)^{2\alpha-1}} \cdot \mathbf{1}_{\alpha \geq 0.5} + \frac{G^{2-4\alpha+4\beta} T^{1-2\alpha+2\beta}}{1-2\alpha} \cdot \mathbf{1}_{\alpha < 0.5} \right),$$

where we used the the Lipschitzness of $y^*(\cdot)$ in the third inequality.

Summarizing all the terms, we finish the proof.

$\square$

