# OpenReview forum: "TiAda: A Time-scale Adaptive Algorithm for Nonconvex Minimax Optimization"
_ICLR.cc/2023/Conference — ICLR 2023 poster_

### Official Review · Reviewer_K2fD · 2022-10-23

**Confidence:** 5
**Correctness:** 4
**Technical Novelty And Significance:** 3
**Empirical Novelty And Significance:** 3
**Recommendation:** 8

**Clarity, Quality, Novelty And Reproducibility:**

(1) How would the algorithm performs if the value of $\delta$ is very small, e.g., at the order of 1e-4 ?
(2) I feel like the additional dependence on $\delta$ is mainly caused by the single loop structure. If we design the algorithm in a nested-loop way, in which we apply adaptive gradient update for both x and y separately, then it seems that we have a good chance to obtain a total sample complexity of $\epsilon^{-4}$, which is better than the rate established in this paper. Can the author discuss a little more about that? If that is the case then the contribution of this paper might not be strong enough.

**Strength And Weaknesses:**

Strength:
(1) It seems to be the first provably efficient adaptive gradient-type algorithm in the non-convex-strongly-concave optimization
(2) The technique proof is solid and the method is very nature and intuitive
(3) This paper provides sufficient empirical evidences

Weakness:
The major weakness is the sample complexity result in the stochastic setting, which is slightly worse than the standard result (standard is $\epsilon^{-4}$ and this paper gives $\epsilon^{-4-\delta}$). It seems that this additional $\delta$ is caused by the single-loop structure and the stepsize scheme thus is hard to improve. I notice that in the experiments the value of $\delta$ is not very small, which means that in theoretical the convergence rate is much worse than SOTA.

**Summary Of The Paper:**

This paper proposes a single-loop adaptive gradient-based algorithm for solving nonconvex-strongly-concave problem with provably efficiency. Previous works only consider directly applying adaptive update scheme to x and y separately thus break the time-scale separation rule which is vital for guaranteeing the convergence. In this paper, the author proposed to adopt a more conservative stepsize for x so that the time-scale separation rule can be followed. The method is simple but effective. The author provide both theoretical and empirical result to verify the effectiveness of their proposed algorithm.

**Summary Of The Review:**

Overall this paper is well-written and easy to follow. The algorithm proposed in this paper is also very interesting and intuitive. The only weakness of this paper is its sample complexity result in the stochastic setting, which could either caused by the nature of the algorithm or the proof technique. I will give borderline accept score for now. If the author can address such an issue later I will consider further increase my score.

---

> ### Author Response · Authors · 2022-11-17
> **To Reviewer K2fD (part 1/2)**
>
>    Thanks for the comments of the reviewer!
>
>    >**Q1: I notice that in the experiments the value of $\\delta$ is not very small, which means that in theoretical the convergence rate is much worse than SOTA. How would the algorithm performs if the value of $\\delta$ is very small, e.g., at the order of 1e-4?**
>
>    As we mentioned in the paper (see the paragraph above Remark 3.4), the convergence of TiAda can be divided in
>    two stages by the time when the desirable time-scale separation is
>    satisfied. Therefore, the total convergence time is affected by not only
>    the rate $\\mathcal{O}(\\epsilon^{-4})$ but also a constant term exponentially
>    related to $1/(\\alpha - \\beta)$.
>    The trade-off between the two stages,
>    which affects the optimal choices of $\\alpha$ and $\\beta$, could be subject
>    to specific tasks. That is why very small $\\delta$ (e.g., in the order of 1e-4) might not be desirable
>    for all cases. For example, we found that the first stage has more impact in
>    the adversarial quadratic test function, where larger $\\alpha - \\beta$ leads to
>    faster convergence (see Figure 5).
>
>    Although the optimal $\\alpha$ and $\\beta$ can be task-related, we argue that
>    TiAda is by design advantageous compared to conventional
>    GDA or GDA with adaptive stepsizes because it is theoretically converging.
>    In addition, we observe that TiAda converges faster than NeAda in our experiments. Because NeAda unconditionally
>    increases  inner loop iterations, it can be a waste of computation if the accuracy in $y$ is already high and lead to slow convergence. This is also one of our motivation to design single-loop algorithm that automatically adjust the time-scale separation.
>    Importantly, we empirically show that even without tuning $\\alpha$ and
>    $\\beta$, TiAda is already faster. Such arbitrary parameter selection is
>    consistent with our goal of making TiAda tuning-free.
>
> (*continued below*)

---

> > ### Author Response · Authors · 2022-11-17
> > **To Reviewer K2fD (part 2/2)**
> >
> >    >**Q2: The major weakness is the sample complexity result in the stochastic setting, which is slightly worse than the standard result (standard is $\\epsilon^{-4}$ and this paper gives $\\epsilon^{-4-\\delta}$). I feel like the additional dependence on $\\delta$
> >  is mainly caused by the single loop structure. If we design the algorithm in a nested-loop way, in which we apply adaptive gradient update for both $x$ and $y$ separately, then it seems that we have a good chance to obtain a total sample complexity of
> >    $\\epsilon^{-4}$, which is better than the rate established in this paper. Can the author discuss a little more about that? If that is the case then the contribution of this paper might not be strong enough.**
> >
> >   The additional $\\delta$ is not caused by the single-loop structure but a price for parameter adaptivity.
> >
> >   In the previous work [1], NeAda is a nested-loop algorithm that updates $y$ in the inner-loop. However, NeAda can still only achieves $\\widetilde{\\mathcal{O}}(\\epsilon^{-(4 + \\delta)})$ sample complexity with AdaGrad as the inner-loop subroutine when the problem parameters are unknown. Specifically, NeAda achieves $\\widetilde{\\mathcal{O}}(\\epsilon^{-4 })$ complexity when the subroutine has a  $\\widetilde{\\mathcal{O}}(T^{-1 })$ convergence rate for solving strongly-concave problems.
> >    However, stochastic AdaGrad can only achieve the $\\widetilde{\\mathcal{O}}(T^{-1 })$ convergence rate when strongly-concavity parameter $\\mu$ is known, and the $\\widetilde{\\mathcal{O}}(T^{-(1 -\\delta)})$ convergence rate when  $\\mu$ is unknown (see Theorem 3.4 and its comments in [1]). TiAda could also achieve $\\mathcal{O}(\\epsilon^{-4})$ complexity if we know the parameters, but it is against our goal of being parameter-agnostic. While both of them achieve near-optimal convergence rates, compared to the nested-loop NeAda, TiAda also enjoys other advantages, e.g., not requiring $\\Omega(\\epsilon^{-2})$ batchsize and being fully parameter-agnostic.
> >
> >    On the other hand, we can consider this $\delta$ in the complexity as the price paid for the adaptivity to the desirable time-scale separation. As showed in Equation (3) from the paper, the effective stepsize ratio is upper bounded by a monotone decreasing sequence if $\\alpha > \\beta$, where $\\alpha$ and $\\beta$ are the powers in the denominators of primal and dual stepsizes, respectively. The difference between $\\alpha$ and $\\beta$  makes this sequence decrease, and we therefore expect the effective stepsize ratio can drop below the desirable time-scale separation. At the same time, the powers $\\alpha$ and $\\beta$ make the stepsize behave as of orders $\\Theta\\left(1/T^\\alpha\\right)$ and $\\Theta\\left(1/T^\\beta\\right)$, while non-adaptive stepsizes for this setting are chosen to be
> >    $\\Theta\\left(1/T^{1/2}\\right)$ for both $x$ and $y$ to achieve the optimal complexity (see
> >    Theorem 3.1 in [2] and Proposition 3 in [3]). When we require $\\alpha$ and $\\beta$  to be different for the sake of adaptivity, their deviation from $1/2$ leads to a complexity of $\mathcal{O}\\left(1/T^\\alpha\\right) + \mathcal{O}\\left(1/T^{1-\\alpha}\\right) + \mathcal{O}\\left(1/T^\\beta\\right) + \mathcal{O}\\left(1/T^{1-\\beta}\\right)$ as in Theorem 3.2.
> >
> >
> >    Single-loop algorithms are preferable in practice, and
> >    we note that designing single-loop AdaGrad-type stepsize for both variables is already technically challenging, since simple combination of GDA and AdaGrad will not even converge when problem parameters are not available as illustrated in the paper.
> >    Moreover, the proving techniques of TiAda are totally different from existing work.
> >
> >
> >    [1] Junchi Yang, Xiang Li, and Niao He. Nest your adaptive algorithm for parameter-agnostic nonconvex minimax optimization. NeurIPS, 2022.
> >
> >    [2] Junchi Yang, Antonio Orvieto, Aurelien Lucchi, and Niao He. Faster single-loop algorithms for minimax optimization without strong concavity. AISTATS, 2022.
> >
> >    [3] Tianyi Chen, Yuejiao Sun, and Wotao Yin. Closing the gap: Tighter analysis of alternating stochastic
> > gradient methods for bilevel problems. NeurIPS 2021.

---

> > > ### Comment · Reviewer_K2fD · 2022-12-14
> > > **Thanks for the reply**
> > >
> > > Thanks author for the detailed explaination. My concern has been addressed so I will raise my score to accept.

---

> > > > ### Author Response · Authors · 2022-12-14
> > > > **Thanks to Reviewer K2fD**
> > > >
> > > > We appreciate the engagement and acknowledgment of the reviewer. Thanks for spending your time carefully reviewing our paper and responding to our clarifications.

---

> ### Author Response · Authors · 2022-12-08
> **The author-reviewer discussion period is ending**
>
> Dear Reviewer K2fD,
>
> We thank the reviewer for the comments and would like to kindly ask if we have addressed your questions about the obtained convergence rate and the source of $\\delta$ in the rate. As the rebuttal period is ending soon, we would appreciate it if the reviewer can give us some feedback and we are happy to make further clarifications.

---

> ### Author Response · Authors · 2022-12-13
> **Has our response addressed your concerns? Sincerely hoping for feedback from Reviewer K2fD**
>
> Dear Reviewer K2fD,
>
> The rebuttal period is ending soon within one day, and we appreciate it if we can have an opportunity to engage with the reviewer. Please let us know whether our response has addressed your concerns.
>
> Best regards,
> Authors

---

### Official Review · Reviewer_Grdi · 2022-10-24

**Confidence:** 3
**Correctness:** 4
**Technical Novelty And Significance:** 2
**Empirical Novelty And Significance:** Not applicable
**Recommendation:** 5

**Clarity, Quality, Novelty And Reproducibility:**

clear to follow

code provided for verification

**Strength And Weaknesses:**

Strength: adaptive stepsizes to nonconvex minimax problems in a parameter-agnostic manner.


Weaknesses: experiment is too simple to verify the algorithm effectiveness.

**Summary Of The Paper:**

In this work, the authors propose a single-loop adaptive GDA algorithm called TiAda for nonconvex minimax optimization that automatically adapts to the time-scale separation. The algorithm is parameter-agnostic and can achieve near-optimal complexities simultaneously in deterministic and stochastic settings of nonconvex strongly-concave minimax problems. The effectiveness of the proposed method is further justified numerically for a number of machine learning applications.

**Summary Of The Review:**

The authors propose a single-loop adaptive GDA algorithm for nonconvex minimax optimization that automatically adapts to the time-scale separation. Only some simple experiment are conducted to verify the algorithm effectiveness.

---

> ### Author Response · Authors · 2022-11-17
> **To Reviewer Grdi**
>
>    Thanks for the comment!
>
>    >**Weaknesses: experiment is too simple to verify the algorithm effectiveness.**
>
>    We emphasize that the focus of our paper is to provide a **theoretically grounded** adaptive method for minimax optimization that is parameter-agnostic and near-optimal for nonconvex-strongly-concave minimax optimization.
>    TiAda is the first single-loop algorithm of this kind.
>    We also respectfully disagree that our experiment is too simple to verify the effectiveness. First,  we use simple test functions (see Figures 1 and 2) only to provide the motivation and intuition behind our algorithm design, as it is convenient to visualize and analytically prove that simple combinations of GDA and adaptive stepsizes — the common practice — can  go wrong, while our algorithm  fixes the issue.  Albeit simple, the effectiveness of our algorithm is already indisputable, in our humble opinion.
>
>    Second,  our experiments include not only simple test functions, but also several complicated tasks with  large datasets.  In the distributional robustness optimization  task, we used convolutional neural networks (CNN) and conducted the experiments on MNIST dataset with a wide range of parameter settings, see Figures 6 and 7. For the generative adversarial network experiments, we used CIFAR-10 dataset and CNNs for generator/discriminator, see Figure 4.
>    While we believe the effectiveness of our method is sufficient justified, we appreciate if the reviewer can
>    provide more concrete suggestions on how we can further improve the numerical experiments in the future.

---

> ### Author Response · Authors · 2022-12-08
> **The author-reviewer discussion period is ending**
>
> Dear Reviewer Grdi,
>
> We thank the reviewer for the comments and would like to kindly ask if we have addressed your concern about the experiments. As the rebuttal period is ending soon, we would appreciate it if the reviewer can give us some feedback and we are happy to make further clarifications.

---

### Official Review · Reviewer_1pAx · 2022-10-25

**Confidence:** 4
**Correctness:** 4
**Technical Novelty And Significance:** 4
**Empirical Novelty And Significance:** 4
**Recommendation:** 8

**Clarity, Quality, Novelty And Reproducibility:**

The contribution and the originality of this work are clearly written in the paper.

**Strength And Weaknesses:**

- S1: The method is parameter agnostic.
- S2: This achieves the optimal complexity for the deterministic case, and the near-optimal complexity for the stochastic case, which improves upon those of NeAda.
- S3: This can be easily generalized to accommodate other existing adaptive schemes.
- W1: Although $\alpha$ and $\beta$ are chosen to be $0.6$ and $0.4$ as a default throughout the experiment, they are hyper-parameters that might ask for tuning in other practical problems.

**Summary Of The Paper:**

Finding an approximate stationary point of a nonconvex(-strongly-concave) minimax problem usually requires the knowledge of problem-dependent parameters, especially the step-size ratio between $x$ and $y$. The most relevant method, named NeAda, is the first parameter-agnostic (adaptive) gradient method for nonconvex minimax problems, but this is double-loop, does not have the optimal complexity, and has several other drawbacks as mentioned in the paper. This paper thus proposes a single-loop parameter-agnostic (two-time-scale) adaptive gradient method, named TiAda, which resolves aforementioned drawbacks of NeAda. There already exist single-loop two-time-scale adaptive gradient methods, but they only work with the knowledge of the problem-dependent parameters. The main new ingredient of the TiAda is having the effective step-size ratio of $x$ and $y$ being upper bounded by a decreasing sequence, making the ratio eventually decrease below the step-size ratio threshold. TiAda (with theoretical result) and other adaptive variants are found to work well in practice.

**Summary Of The Review:**

This paper constructed a single-loop two-time-scale parameter-agnostic adaptive gradient method, named TiAda, for nonconvex  minimax problems, which resolves issues with the first existing parameter-agnostic method, named NeAda. This TiAda method is a simple modification of existing two-time-scale adaptive gradient methods for minimax problems, but it is quite effective both in theory and practice. I believe this work is an important step towards resolving the non-convergence in practical minimax problems.

---

> ### Author Response · Authors · 2022-11-17
> **To Reviewer 1pAx**
>
>    Thanks for the recognition of our work!
>
>    >**W1: Although $\\alpha$ and $\\beta$ are chosen to be 0.6 and 0.4 as a default throughout the experiment, they are hyper-parameters that might ask for tuning in other practical problems.**
>
>
>    We note that TiAda always converges for any hyperparameters $0 < \\alpha < \\beta < 1$ even without tuning, while the existing single-loop methods such as GDA and AdaGrad may diverge. Although the optimal values for $\\alpha$ and $\\beta$ might
>    differ for different applications,  we fix $\alpha$ and $\beta$ throughout our experiments and observe that TiAda consistently outperforms others, which conforms to our goal
>    of designing a parameter-agnostic algorithm. In Figure 2(h) and Figure 5, we also provide some illustration on the performance under different choices of $\alpha$ and $\beta$. Of course, TiAda could have even better performance if we tune these hyperparameters, but it is against our purpose. Lastly, hyperparameters  commonly exist in adaptive methods, e.g., initial stepsize in AdaGrad and momentum parameters in Adam. Similar to AdaGrad and Adam in minimization problems, TiAda achieves fast convergence in minimax optimization even when we use default hyperparameters.

---

### Official Review · Reviewer_hniQ · 2022-10-26

**Confidence:** 3
**Correctness:** 3
**Technical Novelty And Significance:** 3
**Empirical Novelty And Significance:** 3
**Recommendation:** 6

**Clarity, Quality, Novelty And Reproducibility:**

The paper is well-written and easy to follow. The intuition behind the algorithm is clear. While the algorithm has some features of prior work, the improvements made by the proposed method are somewhat novel.


**Strength And Weaknesses:**

**Strengths:**

1. The proposed method improves upon the main related prior work such as similar implementation in deterministic and stochastic cases (agnostic to the noise level in the gradient). It does not need complex subroutines in the inner loop update for termination.
2. The work provides extensive validation of the proposed method against related baselines, in their experimental setting, and provides an ablation over some proposed algorithm-specific parameters.
3. The intuition of the work is clear and the paper is easy to follow.

**Weaknesses:**

1. The rate obtained in Theorem 3.2 is worse compared to state-of-the-art Na-Ada. Could you explain the reason behind this worse rate?
2. It is unclear how the list of values for r is chosen for results in Figure 2. What is the reason for choosing different orders of ratios for the two other experiments? It seems like NeAda would perform better for smaller choices of r. Furthermore, how was $\eta_x$ (or $\eta_y$) chosen for these experiments?
3. It is not clear why Na-Ada is not included in GAN simulation in Figure 4.
4. Some typos in the text (e.g. definition in Assumption 3.2).


**Summary Of The Paper:**

The authors propose an algorithm named TiAda which is a time-scale adaptive algorithm for non-convex-strongly-convex (NC-SC) minimax problems. The algorithm is a single loop and problem-specific parameter agnostic one, which are improvements over the related prior work. The authors provide insight into the design of the algorithm, theoretically analyze the algorithm and empirically validate its usefulness of the algorithm.  The authors also provide some generalization of TiAda to other adaptive methods and empirically validate their usefulness against related baselines.


**Summary Of The Review:**

The paper proposes a parameter-agnostic adaptive algorithm for solving NC-SC minimax problem. The authors have identified a gap in the literature in this regard and have provided a somewhat novel contribution. The authors provide the intuition as to why their solution addresses the prevailing issues, and theoretically and empirically justify their claims. The algorithm seems to be robust to the hyperparameter choices required by the algorithm, as per the results shown by the authors. The empirical results provided by the authors suggest the proposed methods outperform related baselines. There are some issues regarding the experimental setup in some of the empirical results, as mentioned above under “Weaknesses”. Furthermore, the theoretical results do not improve the prior work in the stochastic setting.

---

> ### Author Response · Authors · 2022-11-17
> **To Reviewer hniQ (part 1/2)**
>
>    Thanks for the comments!
>
>    > **Q1: The rate obtained in Theorem 3.2 is worse compared to state-of-the-art NeAda. Could you explain the reason behind this worse rate?**
>
> In the deterministic setting,  TiAda achieves the optimal convergence rate of $\\mathcal{O}\\left(\\epsilon^{-2}\\right)$ and improves over the state-of-the-art NeAda by shaving off an extra logarithmic term.  In the stochastic setting, TiAda attains $\\mathcal{O}\\left(\\epsilon^{-(4+\\delta)}\\right)$ sample complexity for arbitrarily small $\\delta>0$, which can be made arbitrarily close to the optimal rate $\\mathcal{O}\\left(\\epsilon^{-4}\\right)$.  NeAda achieves $\\widetilde{\\mathcal{O}}(\epsilon^{-4 })$ complexity (with extra logarithmic term) only  when the inner-loop subroutine has a  $\\widetilde{\\mathcal{O}}(T^{-1 })$ convergence rate for solving strongly-concave problems. If the subroutine is chosen to be AdaGrad (same as what is adopted in our TiAda),  NeAda can only achieve $\\widetilde{\\mathcal{O}}(\epsilon^{-(4 + \\delta)})$ sample complexity (see Theorem 3.4 and its comments in [1]), which is similar to ours but with extra logarithmic term.  Note that stochastic AdaGrad can only achieve the $\\widetilde{\\mathcal{O}}(T^{-1 })$ convergence rate when strongly-concavity parameter $\\mu$ is known, and the $\\widetilde{\\mathcal{O}}(T^{-(1 -\\delta)})$ convergence rate when  $\\mu$ is unknown. The additional $\delta$ in the complexity can be viewed as the price paid for the adaptivity, yet it can be arbitrarily close to zero.  In summary, the rate of TiAda is not necessarily worse than NeAda; they are quite comparable and both are near-optimal, while TiAda also enjoys other advantages, e.g., not requiring $\\Omega(\\epsilon^{-2})$ batchsize and being fully parameter-agnostic.
>
>
>
>    > **Q2: It is unclear how the list of values for $r$ is chosen for results in
>    Figure 2. What is the reason for choosing different orders of ratios for the
>    two other experiments? It seems like NeAda would perform better for smaller
>    choices of $r$. Furthermore, how was $\\eta\_x$ (or $\\eta\_y$)
>    chosen for these experiments?**
>
>   For the fair comparison, we use the same $r$ values as in Figures 1 and 2 of [1], where they choose different $r$
>   that are above, near and below the time-scale separation threshold in order to test the effectiveness of the proposed algorithm.
> The two experiments correspond to different test functions, quadratic and McCormick functions respectively, thus their theoretical thresholds are different;  therefore, we pick different orders of ratios.
>
>    In Figure 2, TiAda outperforms NeAda with all ratios. If we look at
>    the performance of a single algorithm across different ratios,
>    they all converge faster when
>    $r$ is smaller, because
>    smaller initial stepsize ratio that are near or below the threshold will benefit the convergence. This also illustrates that TiAda is most robust to different initial stepsize ratios among these algorithms.
>
>
>    We choose $\\eta\_x$ and $\\eta\_y$ the same as [1] in our experiments with test functions. For the real-world applications, we exhibit the results with different combinations of $\\eta\_x$ and $\\eta\_y$, e.g., Figure 7 in the appendix. TiAda has consistently good performance with different initial stepsizes.
>
>
> (*continued below*)

---

> > ### Author Response · Authors · 2022-11-17
> > **To Reviewer hniQ (part 2/2)**
> >
> >
> >    > **Q3: It is not clear why NeAda is not included in GAN simulation in Figure 4.**
> >
> >   We did try the GAN experiment on NeAda, and observed that it performed poorly in this task.    For NeAda,
> >    the number of inner loop iterations increases gradually, and as a result,
> >    after the early stage of training, the discriminator (or critic) is trained extremely
> >    well for each generator update. While a good approximation of the inner maximization problem is
> >    desirable for NC-SC problems, it is less known in the NC-NC setting.
> >    For example, too strong discriminator can lead to vanishing gradients for
> >    the generator [2]. Also, the relatively aggressive
> >    increase of inner loop iterations in NeAda might cause a waste of computation if the maximization problem is already well solved and lead to slow convergence.
> >    This is also one of our motivations to introduce single-loop TiAda that adjusts stepsize ratio on the fly.
> >
> > Nevertheless, it is worthy emphasizing that GANs are inherently nonconvex-nonconcave, and there is no theory supporting that NeAda should work well on such problems, so does our TiAda.  Running such experiments, unless to an extensive level, may not offer more meaningful insights than we have shown in other experiments. However, the GAN simulation may show some potential of TiAda in a more generalized setting.
> >
> >
> >
> >
> >    > **Q4: Some typos in the text (e.g. definition in Assumption 3.2).**
> >
> >    Thanks for pointing this out! We fixed this in the revision.
> >
> >
> >    **References**
> >
> >
> >    [1] Junchi Yang, Xiang Li, and Niao He. Nest your adaptive algorithm for parameter-agnostic nonconvex minimax optimization. NeurIPS, 2022.
> >
> >    [2] Martin Arjovsky and Leon Bottou. Towards Principled Methods for Training Generative Adversarial Networks. In *ICLR*, 2017.

---

> ### Author Response · Authors · 2022-12-08
> **The author-reviewer discussion period is ending**
>
> Dear Reviewer hniQ,
>
> We thank the reviewer for the comments and would like to kindly ask if we have addressed your questions on the obtained convergence rate and the experimental details. As the rebuttal period is ending soon, we would appreciate it if the reviewer can give us some feedback and we are happy to make further clarifications.

---

### Decision · Program_Chairs · 2023-01-20

**Decision:**

Accept: poster

**Justification For Why Not Higher Score:**

- The results are not really surprising.

**Justification For Why Not Lower Score:**

- The results achieve the optimal complexity for the deterministic case and the near-optimal complexity for the stochastic case.
- Moreover, the work provides extensive validation of the proposed method against related baselines, in their experimental setting, and provides an ablation over some proposed algorithm-specific parameters.

**Metareview: Summary, Strengths And Weaknesses:**

The paper proposes a single-loop adaptive gradient-based algorithm for solving nonconvex-strongly-concave problem with provably efficiency. The results achieve the optimal complexity for the deterministic case and the near-optimal complexity for the stochastic case. Moreover, the work provides extensive validation of the proposed method against related baselines, in their experimental setting, and provides an ablation over some proposed algorithm-specific parameters.

The authors have also addressed the concerns from the reviewers properly and most of the reviewers are happy to support its publication. Please consider the comments and suggestions from the reviewers and incorporate them into the final version.


**Note From Pc:**

if the above contains the word "oral" or "spotlight" please see: "oral" presentation means -> notable-top-5% and "spotlight" means -> notable-top-25%. As stated in our emails, we are disassociating presentation type from AC recommendations